# Royal Jelly as an Intelligent Anti-Aging Agent—A Focus on Cognitive Aging and Alzheimer’s Disease: A Review

**DOI:** 10.3390/antiox9100937

**Published:** 2020-09-29

**Authors:** Amira Mohammed Ali, Hiroshi Kunugi

**Affiliations:** 1Department of Mental Disorder Research, National Institute of Neuroscience, National Center of Neurology and Psychiatry, Tokyo 187-0031, Japan; hkunugi@ncnp.go.jp; 2Department of Psychiatric Nursing and Mental Health, Faculty of Nursing, Alexandria University, Alexandria 21527, Egypt; 3Department of Psychiatry, Teikyo University School of Medicine, Tokyo 173-8605, Japan

**Keywords:** Alzheimer’s disease, neurodegenerative disorders, aging, alternative therapy, apitherapy, amyloid-β, cognitive impairment, dementia, mitochondrial dysfunction, oxidative stress, neuroinflammation, gut-brain axis, royal jelly

## Abstract

The astronomical increase of the world’s aged population is associated with the increased prevalence of neurodegenerative diseases, heightened disability, and extremely high costs of care. Alzheimer’s Disease (AD) is a widespread, age-related, multifactorial neurodegenerative disease that has enormous social and financial drawbacks worldwide. The unsatisfactory outcomes of available AD pharmacotherapy necessitate the search for alternative natural resources that can target the various underlying mechanisms of AD pathology and reduce disease occurrence and/or progression. Royal jelly (RJ) is the main food of bee queens; it contributes to their fertility, long lifespan, and memory performance. It represents a potent nutraceutical with various pharmacological properties, and has been used in a number of preclinical studies to target AD and age-related cognitive deterioration. To understand the mechanisms through which RJ affects cognitive performance both in natural aging and AD, we reviewed the literature, elaborating on the metabolic, molecular, and cellular mechanisms that mediate its anti-AD effects. Preclinical findings revealed that RJ acts as a multidomain cognitive enhancer that can restore cognitive performance in aged and AD models. It promotes brain cell survival and function by targeting multiple adversities in the neuronal microenvironment such as inflammation, oxidative stress, mitochondrial alterations, impaired proteostasis, amyloid-β toxicity, Ca excitotoxicity, and bioenergetic challenges. Human trials using RJ in AD are limited in quantity and quality. Here, the limitations of RJ-based treatment strategies are discussed, and directions for future studies examining the effect of RJ in cognitively impaired subjects are noted.

## 1. An Overview of Cognitive Aging

The world has witnessed an astronomical increase in the aged population, both in developed and developing countries. Unfortunately, longevity comes at a high price. Aging is associated with progressive deterioration of overall homeostasis and a decline of cognitive, visual, hearing, and muscular functions, as well as sleep. These functional alterations lead to a trail of burdensome problems including fatigue, mood dysregulation, overall physical dysfunction, low quality of life, low life-expectancy, high disability and dependency, and eventually institutionalization [1,2,3,4,5,6].

Evidence indicates that aged brains undergo several pathological changes: higher metabolic stress, reduced neurogenesis, increased synaptic aberrations, immune dysregulation (high expression of inflammatory markers), and low expression of neuroprotective factors. Altered brain physiology, along with disturbances in the function and synchronization of the circadian system, significantly increase the prevalence of neurodegeneration, neurobehavioral deficits, and cognitive aging [7,8]. Research documents that normal cognitive aging involves synaptic changes and decreased neuronal plasticity of the cortex and hippocampus, which directly accelerate declines in memory, reasoning, and speed; these alterations may start from early adulthood [9,10].

Chronic neuroinflammation is one of the main pathologies that contribute to age-related cognitive decline. In particular, microglia (the resident immune cells of the brain) are persistently activated in response to neuroinflammation, leading to cytotoxic effects and pathologic changes in cortical white matter [11], which involve protein nitration, white mater lesions (signal abnormalities), and myelin loss. White matter tracts play a major role in learning and information processing, whereas age-related alterations of white matter represent a main source of cognitive deterioration among the elderly. Indeed, the acceleration of age-dependent microglial activation and proliferation in human white matter and its associated disruption of white matter integrity can be detected early in middle-age (around the age 50 years) [12]. In addition, cerebral arterial stiffness increases with age, and is associated with the formation of white matter lesions and cognitive decline in normal aging [13]. Interestingly, the term “normal cognitive aging” can be misleading, given that researchers have identified discrete cognitive phenotypes that reflect nonoverlapping vulnerabilities for cognitive decline (e.g., speed and memory) in high-functioning adults. Therapeutic targets within the “normal” spectrum can be identified early through suitable biomarkers [14]. Altogether, increased aging remains the main underlying etiology of cognitive deterioration in the elderly population [14,15].

## 2. Overview of Alzheimer’s Disease

Dementia is a clinical syndrome that manifests mainly by progressive and irreversible deficits in cognitive performance, which lead to complete dependence for activities of daily living (ADL) [5,16]. Worldwide, dementia represents the second most prevalent neurological disorder (around 50 million cases in 2015) after different types of headache [17]. Alzheimer’s disease (AD) accounts for 70 to 80% of dementia cases. Currently, 36 million people suffer from AD. Around 5 million new cases develop every year, and it is estimated that the AD population will reach 115 million cases by 2050, given the continuous increases in life expectancy [5,18]. AD is the sixth leading cause of death in the USA, and its annual cost of care exceeds $232 billion, making it the third most burdensome disease after cancer and coronary heart disease. It represents a massive form of disability that significantly impairs all aspects of life, including lower life expectancy, poor quality of life, and high levels of institutionalization. In addition, family caregivers of AD patients are vulnerable to serious mental and physical health problems due to burnout and distress [5,19].

Few people get diagnosed and treated in the early stages of AD, despite the fact that the disease can be clinically diagnosed by physical and neurological examination, and its symptoms may be effectively managed in the early stages. This is because the brain pathology of AD may precede the clinical diagnosis of dementia by up to 20 years [20,21,22]. In addition, the course of the disease is highly diverse. AD comprises three stages based on the level of cognitive and functional impairments: (1) a preclinical stage characterized by normal cognitive performance, which can last up to 10 years, (2) a prodromal stage characterized by mild cognitive impairment (MCI), which can last up to 4 years, and (3) dementia, which is characterized by manifest functional impairments, which may last longer than 6 years [14,22]. Moreover, a lack of agreement among AD specialists on the terminology related to clinically “normal” cognitive aging and dementia may cause patients with preclinical AD to miss the chance of getting promptly diagnosed and treated [23,24].

In addition to cognitive deterioration, the main characteristic symptom of AD, numerous other symptoms develop such as language disturbances, functional impairments in ADL, and a wide range of neuropsychiatric symptoms, e.g., depression, irritability, anxiety, agitation, apathy/indifference, delusions, and hallucinations. Evidence indicates that depression and irritability represent early AD symptoms that exist alongside MCI before AD diagnosis [25]. The course of the disease begins with mild symptoms that gradually get more severe [26]. Symptoms associated with AD heighten caregivers’ burden and increase rates of institutionalization and mortality [25].

## 3. The Mechanism Underlying AD Development

Understanding the pathophysiology of AD is necessary for early detection and the development of specific effective treatments. Despite extensive investigations of AD, the exact cause remains unclear. However, AD researchers came to a consensus on the main features of AD pathogenesis: progressive buildup of the neurotoxic oligomers of beta-amyloid (Aβ) protein fragments to form insoluble senile plaques (SPs) outside neurons. The amyloidogenic pathway is considered the main molecular event contributing to the accumulation of Aβ. The rate-limiting step in the process of Aβ production involves proteolytic cleavage of amyloid precursor protein (APP) by β-secretase (beta-site APP cleaving enzyme 1, BACE1), which results in the production of part-soluble APP peptide-b and C-terminal APP fragment-b, which then get further cleaved by γ-secretase to produce hydrophobic Aβ polypeptides [27,28].

### 3.1. Role of the Immune System in Alzheimer’s Disease

AD pathology in 99% of cases involves multiple interactions between environmental, lifestyle, and genetic factors. Genetics constitute 53% of total phenotypic variance [29]. A current wide-scale meta-analysis of genome-wide association studies has identified 215 risk-increasing genes of AD. Most of these genes are related to body tissues involved in the immune system: whole blood, spleen, liver, and microglia [30]. Accumulating evidence confirms the core casual involvement of the immune system and chronic neuroinflammation in the pathology of age-related neurodegenerative diseases, such as AD [11,31]. Aged brains that undergo inflammation develop a ‘sensitized’ or ‘primed’ phenotype, mainly because microglia exhibit dystrophic morphology, increased production of pro-inflammatory molecules, and diminution of neuroprotective factors [32]. A sensitized phenotype of the aged brain is highly vulnerable to secondary insults such as infections and psychological stress. Inflammatory cytokines foster the transcriptional upregulation of β-secretase and APP and increase Aβ aggregation, which contribute to the characteristic neuropathologic substrate of AD—Aβ plaques [7,11,33].

Neuroinflammation synergizes the expression of the genes involved in AD development in aged brains. In this respect, astrocytes play a major role in the neuroinflammatory and neurodegenerative processes underlying AD. They represent the main site where genes associated with AD are expressed, such as the apolipoprotein E (APOE) gene [34]. APOE-ε4, the main APOE isomer, is strongly associated with the onset of late-onset familial and sporadic AD [35]. In mice, APOE-ε4 is associated with low spontaneous excitatory postsynaptic currents in the amygdala in middle age and high excitatory activity in old age, with multiple long-term hippocampal alterations related to metabotropic glutamatergic receptors, i.e., the extrasynaptic *N*-methyl-d-aspartate receptor (NMDAR)-dependent signaling pathway. Inheritance of APOE-ε4 in humans is associated with alterations of the brain structure and function, which occur very early before the onset of AD, e.g., behavioral deficits and reduced glucose metabolism in the temporal cortex and parahippocampal gyrus occur in young and middle-age individuals. In old age, APOE-ε4 is associated with heightened brain atrophy in the medial temporal lobe and accelerated Aβ deposition [21]. Moreover, carriers of APOE-ε4, individuals in early stages of AD, and cognitively normal people with Aβ aggregation demonstrate hyperactivity or dysregulated activation of the default-mode network during hippocampal memory-encoding. Experimentally, these dysregulations occur even before the formation of Aβ lesions [36]. Though the exact mechanism through which APOE contributes to neurodegeneration is not well-understood, current research indicates that APOE-lipoproteins bind to various cell-surface receptors and interfere with processes essential for the generation and clearance of Aβ and tau, such as the transport of brain lipid (e.g., lipophilic Aβ peptide), neuronal signaling, mitochondrial function, and glucose metabolism [37]. Furthermore, overproduction of APOE by activated microglia might exacerbate neuroinflammation, leading to sporadic brain neurodegeneration [31]. In line with this, our research group demonstrated increased APOE protein levels in the cerebrospinal fluid of people with APOE-ε4 allele, compared to those without the allele [38].

### 3.2. Role of the Gut-Brain Axis in AD Pathogenesis

From another angle, emerging knowledge attributes the origin of the inflammatory processes underlying AD to microbiome alterations [39]. Various external factors such as imbalanced diet (high in fat and low in fiber) and the ingestion of toxins and pathogens cause propagation of enterotoxigenic bacterial strains such as Gram-negative bacilli *Bacteroides fragilis* and *Escherichia* (*E.*) *coli*. This imbalance is associated with a reduction of beneficial bacteria such as *Lactobacillus johnsonii* [39,40,41]. Under imbalanced gut conditions, the symbiotic interactions of the gut microbiome that are essential for human physiological functions get disrupted, allowing enterotoxigenic bacteria to cause virulent aberrations and inflammation of the gastrointestinal (GI) wall [42]. Consequently, the production and availability of neuroactive molecules produced by enteric neurons (e.g., serotonin) deteriorate [39,41,43]. Furthermore, inflamed gut undergoes local activation of inducible nitric oxide synthase (iNOS, also known as NOS2) signaling and increased expression of gut nitric oxide (NO). Local activation of inflammatory pathways follows iNOS activation, leading to increased production of cytokines, which further accelerates GI inflammation and dysbiosis/leaky gut. GI dysbiosis enables endobacteria and their toxins, i.e., potent pro-inflammatory neurotoxins (e.g., lipopolysaccharide (LPS)), to get into the blood stream and induce systemic inflammation and oxidative stress. All these events lead to pathological clotting, leakage of the blood brain barrier (BBB), suppression of neurotrophins, and the initiation of neuroinflammatory processes that lead to the accumulation of Aβ [39,40,41,43] (Figure 1).

As illustrated by the yellow arrows, the ingestion of pathogens, toxic substances, and improper diet (low fiber and high fat) disrupt the gut microbiota. Microbiota imbalance causes the propagation of highly toxic endobacteria such as Gram-negative bacteria and a reduction in numbers of beneficial bacteria such as lactic acid bacteria and bifidobacteria. Endobacteria toxins cause local inflammation and oxidative stress, resulting in aberrations and injuries of the gastrointestinal wall, which allow the passage of endobacteria and their toxins into the blood stream, as illustrated by the red arrows. The persistent gastrointestinal leak causes continuous activation of the immune system. The activation of pathways of inflammation and oxidative stress such as NF-κB and iNOS results in high production of inflammatory cytokines, free radicals, adhesion factors, and advanced glycation end-products. The latter form as a result of nonenzymatic glycation and the oxidation of lipids, long-lived proteins, and nucleic acids. They bind to RAGE and other cell-surface receptors or cross-link with other proteins to induce a vicious cycle of oxidative stress and inflammation. Bacterial toxins, along with cytokines, ROS, and activated adhesion factors attack the tight junction proteins of the BBB, leading to their degradation and permeability of the BBB, which allows the passage of toxic solutes (e.g., Aβ) from the blood stream to the brain tissue. Glial cells (microglia, astrocytes, and oligodendrocytes—star-shape-like cells colored in brown on Figure 1) get activated in order to protect neurons against toxic Aβ and other products of inflammatory and oxidative responses. Glial activation phenotype develops as a result of the depletion of resources of these cells and their failure to counteract constant inflammation, which results in a cycle of self-perpetuating neuroinflammation and neuronal toxicity.

Of interest, rodents fed fecal *E. coli* from rats with colitis exhibited significant memory and learning impairment, along with changes in the composition of gut microbiota and elevation of LPS levels in the feces, blood, and brain [40]. Evidence reveals that *E. coli* can form extracellular functional amyloid or amyloid-like structures [44,45]. Evolving knowledge indicates that all human brains and the erythrocytes of AD patients contain bacterial-encoded 16S rRNA, which is largely (i.e., more than 70%) Gram-negative bacteria [41,46]. In the meantime, iron dysregulation—as indicated by exceptionally high serum ferritin levels—is common in AD patients; it contributes to oxidative stress and causes virulent reactivation of dormant blood and tissue microbiome [39]. *E. coli* bacterial lipoproteins act as agonists of toll-like receptor (TLR) 2 and nucleotide-binding domain and leucine-rich repeat containing protein 3 (NLRP3); they induce inflammation and increase cytokine production through their interaction with recombinant human Serum Amyloid A1 (hSAA1) [47]. In addition, various levels of bacterial LPS have been detected in blood samples and brain lysates from the superior temporal lobe neocortex and hippocampus of AD patients. Meanwhile, levels of hippocampal LPS in advanced AD cases can be up to 26-fold higher than in cognitively normal age-matched controls [41,48]. LPS localizes in high levels with Aβ1-40/42 in Aβ lesions, around cerebral vessels, and inside oligodendrocytes and neurons in the brain of AD patients [49]. LPS contributes to myelin injury and white matter hyperintensities, a main pathophysiological feature that is associated with the severity of neurodegeneration and cognitive decline in AD. In addition, AD patients exhibit extremely high blood titer of autoantibodies against various myelin proteins, and myelin damage precedes tau pathology and amyloid aggregation in experimental models of AD. Meanwhile, focal loss of myelin and oligodendrocytes occurs within and around Aβ lesions in familial and sporadic AD [50]. Mechanistically, once bacterial LPS crosses the BBB, it targets glial cells (oligodendrocytes) in the white and grey matter by activating leukocyte and glial TLR4-CD14/TLR2 and low-density lipoprotein receptor-related protein 1 (LRP-1). The latter regulates lipid metabolism and transports Aβ out of the brain. LPS binding to these receptors initiates intracellular signaling cascades associated with mitochondrial oxidative stress (e.g., iNOS) and inflammation (e.g., nuclear factor-kappa B; NF-κB) [51,52]. Cellular failure to adapt to the continuously increased production of ROS and proinflammatory cytokines eventually causes serious injuries to oligodendrocytes, degrades myelin proteins, e.g., myelin basic protein (MBP), and stimulates the aggregation of Aβ, because Aβ clearance efficiency of LRP-1 diminishes. Moreover, Aβ1–42 directly associates with LRP-1 and binds to degraded MBP, which accelerates plaque formation. In the meantime, leaky gut and leaky BBB promote chronic LPS intrusion, cellular damage, and Aβ1–42-induced agonism of TLR4 receptors. These events generate a vicious cycle of incorrectly modulated glial activation phenotype that supports progressive neurotoxicity [51,53].

### 3.3. Multiple Medical Conditions Contribute to AD by Enhancing Neuroinflammation

The diminution of sex hormones that occurs with aging is associated with memory and cognitive changes. In particular, the drop in estrogen levels that occurs after menopause aggravates neuroinflammation through the activation of inflammatory pathways such as NF-κB. Accordingly, estrogen deficiency stimulates multiple autonomic nervous changes including Aβ accumulation, resulting in memory impairment and increased susceptibility to AD [24,54]. In addition, loss of bone mass is common in AD patients; it usually occurs as a direct effect of estrogen deficiency [55]. The literature documents sex differences in the prevalence, cerebral pathology, and molecular mechanisms of AD. Compared with male *APP/PS1/tau triple-transgenic* AD mice, AD female mice exhibit more prominent Aβ plaques, neurofibrillary tangles, neuroinflammation, spatial cognitive deficits, and dysregulation of hippocampal cyclic adenosine monophosphate (cAMP)-response element (CRE)-binding protein (CREB) signaling. These differences are likely to be induced by estrogen deficiency in female mice [56].

Advanced aging is associated with multiple systemic failures such as reduced vascular elasticity, including cerebral blood vessels. Emerging knowledge emphasizes the fact that APOE and age-related vascular pathologies such as arteriosclerosis lead to failure of drainage of Aβ and soluble proteins from perivenous spaces [57]. Cerebral artery pathology unrelated to Aβ, which occurs prior to AD, contributes to the impairment of downstream arterioles in AD. Cerebral amyloid angiopathy develops as a result of Aβ deposition on the inner walls of blood vessel, resulting in poorer Aβ drainage [58], the formation of lesions in white matter, and greater a degree of cognitive decline [13,58].

Several lines of evidence suggest that the pathogenetic processes of AD, mainly neuroinflammation, are initiated by various metabolic dysfunctions, e.g., insulin resistance, nutritional problems (e.g., vitamin D deficiency), and hormonal abnormalities e.g., thyroid dysfunction [5,59,60]. Investigations of neural exosomes and nanosomes revealed that AD patients exhibit metabolic disturbances many years before they develop AD [61]. Dysregulated insulin signaling is one of the main metabolic dysfunction that contributes to AD pathology [62]. Insulin is one of the hormones that affect every single cell in the body, and the insulin/insulin-like growth factor (IIS) signaling pathway interacts with other pathways and affects their functioning [63]. It plays a major role in cellular growth, autophagy, energy utilization, mitochondrial function, management of oxidative stress, synaptic plasticity, and cognitive function [62]. Longitudinal and autopsy studies suggest an important contribution of vascular mechanisms (small vessel disease) to cognitive decline in AD associated with type 2 diabetes mellitus [60]. Hyperglycemia induces vascular permeability by upregulating hypoxia inducible factor-1α (HIF-1α), which subsequently activates vascular endothelial growth factor (VEGF) [64]. Hyperglycemia also amplifies inflammatory and oxidative processes by contributing to the production of advanced glycation end-products (AGEs) through nonenzymatic glycation and oxidation of lipids, long-lived proteins, and nucleic acids. AGEs are heterogenous compounds that bind to the receptor for advanced glycation end products (RAGE) and other cell-surface receptors, or cross-link with other proteins to induce oxidative stress and inflammation [33,64]. Meanwhile treatments that resensitize brain insulin signaling have been shown to improve cognitive function in AD patients [62]. In the same way, hypertension reduces brain reserve and contributes to amyloid-dependent pathway signaling, which facilitates Aβ and tau deposition and related neuronal injury, especially in APOE-ε4 carriers, leading to AD development [65]. A recent systematic review shows that the use of any effective antihypertensive medication decreases the incidence of AD in hypertensive patients [66].

Hypercholesterolemia represents another common deleterious age-related health problem which is strongly linked to hypertension [67]. Biophysical properties and signaling of the biomembrane in both physiological and pathological processes are affected by asymmetries of lipid distribution: condensed cholesterol localizes mostly in the external hemilayers in raft domains/lipid-ordered microdomains along with saturated phosphatidyl lipids and sphingolipids [68]. A considerable number of AD genes are related to liver function, denoting a prominent role of lipids in the pathogenesis of AD [30]. High cholesterol levels and dysfunctional lipid metabolism in the brain, which is regulated by APOE, promote the pathogenesis of AD by altering the vascular integrity of the brain and the immune responses of glial cells [30,69,70]. The accumulation of Aβ particles in the cell membrane mostly occurs in raft domains, the cleavage location of the precursor APP by β- and γ- secretase [68]. Human extracellular Aβ fibrils contain large amounts of cholesterol rich lipids. The region of residues 22–35 in the Aβ peptide has been identified as a potential cholesterol binding site where cholesterol vesicles accelerate the primary nucleation of Aβ42 by up to 20-fold via a heterogeneous nucleation pathway [69,70].

AD is a multifactorial disease; several other risk factors with detrimental effects have been proposed such as hypothyroidism, chronic stress (e.g., income inequality), improper diet (e.g., high fat and low fiber), and pollution (e.g., heavy metals) [5,15,41]. These factors affect the homeostasis of the whole body, including the gut-brain axis, leading to aging-like sensitization of microglia and heightened reactivity to secondary insults [7]. As a result, most physiological processes in aging deteriorate, which synergizes AD-related genetic interactions [5,15]. The resulting alterations entail the activation of various pathological pathways that involve the depletion of endogenous antioxidants, oxidative stress, neuroinflammation, dysfunctional proteostasis, expression of proapoptotic proteins leading to mitochondrial breakdown, depletion of neurotrophins, glutamatergic neurotoxicity, synaptic dysfunction, and neuron loss [71,72,73].

### 3.4. Amyloid Pathology Promotes Neurodegeneration by Altering Cellular Structure and Function

Monomers and oligomers of the Aβ peptide are highly disordered, and they disrupt the homeostasis of cellular elements including bio-metals [71]. The mechanism and kinetics through which Aβ aggregates depend on its interaction with transition metal ions such as copper, iron, and zinc [74]. Aβ interacts with zinc (Zn) ions, which exist in large amounts in SPs in AD patients. ZnAβ oligomers adopt the same β-sheet structure as in Aβ fibrils. However, they exhibit a greater potential to accelerate hippocampal microglia activation. Cell viability and cytotoxicity assays show that ZnAβ oligomers are more cytotoxic than Aβ oligomers [75]. Zn is involved in the regulation of multiple AD-related metabolic processes such as hormonal signaling, insulin desensitization, and proteolytic activities. Being trapped inside Aβ plaques, a sort of Zn deficiency occurs, which hastens metabolic, genetic, and epigenetic alterations that contribute to a widespread pathology in the cortex. Thus, this pathology results in early nonamnestic features such as dyscalculia and aphasia [61]. Copper and iron are core redox active transition metals that interact with Aβ to facilitate the electron transfer necessary for the generation of ROS and reactive nitrogen species (RNS) in a reaction that requires tyrosine 10 [76]. Hydrogen peroxide (H_2_O_2_), one of the most prominent free radicals produced in AD brains due to oxidative stress, is an uncharged, stable, and freely diffusible ROS. H_2_O_2_ amplifies Aβ neurotoxicity by interacting with iron and copper to generate highly toxic ROS that accelerate the buildup of inflammatory cytokines and attract activated microglia to encircle Aβ plaques, causing more ROS/H_2_O_2_ production and leading to drastic damages of cellular proteins, lipids, and DNA, and eventually neuronal loss [77,78] (Figure 2).

On one side, Aβ interacts with transition metal ions such as copper, iron, and zinc, leading to the formation of highly toxic Aβ oligomers and high production of hydrogen peroxide (H_2_O_2_), a highly toxic ROS. H_2_O_2_ amplifies Aβ neurotoxicity by interacting with iron and copper to generate highly toxic ROS that accelerate the buildup of inflammatory cytokines and attract activated microglia to encircle Aβ plaques, causing more ROS/H_2_O_2_ production. These events ultimately lead to drastic damages of cellular proteins, lipids, and DNA. On the other side, Aβ activates the phosphorylation of various tau proteins via a mechanism that involves alteration of the ER by oxidative stress. ER is the main region involved in protein folding and secretion. ER stress activates UPR signaling. UPR contextually regulates protein misfolding and neurodegeneration in Alzheimer’s disease. MAP-2, the main protein in neuronal axon, is a key tau protein that is affected by Aβ activity. Hyperphosphorylation of MAP-2 leads to axonal collapse, dendrite instability, degeneration of synaptic spines, and disrupted signal transduction.

Oxidative stress alters endoplasmic reticulum (ER), the main region involved in protein folding and secretion. ER stress activates unfolded protein response (UPR) signaling pathway, which regulates protein misfolding and neurodegeneration. UPR contextually regulates protein misfolding and neurodegeneration in AD [72]. In particular, the phosphorylation of protein kinase RNA like ER kinase (PERK), a stress-responsive transmembrane protein embodied in UPR, leads to the activation of several signaling cascades that regulate redox proteins, cholesterol metabolism, and genes of foldases and chaperones [79]. In this respect, the formation of Aβ plaques is significantly associated with proteotoxicity and tauopathy, i.e., protein misfolding, hyperphosphorylation and nucleation of axonal tau proteins, resulting in a wide spread of twisted strands of the tau protein and the formation of neurofibrillary tangles (NFTs) inside neurons, particularly in neocortical regions [20,23]. However, soluble tau can also be neurotoxic. In addition, the hyperphosphorylation of some tau proteins such as microtubule-associated protein 2 (MAP-2) contributes to neuronal death. MAP-2 is a cytoskeleton protein that regulates dendrite branching, microtubule assembly, and synaptic signal transduction. The hyperphosphorylation of MAP-2 leads to microtubule collapse, dendrite instability, axon degeneration, and dysfunctional axoplasmic transport. As a result, the synthesis, transport, release, and uptake of neurotransmitters gets disrupted. Such interactions promote neurotoxicity, aggravate tau and Aβ deposition, and ultimately lead to more cellular lesions and sporadic neurodegeneration [73,76,80]. Finally, progressive damage and death of neurons in several areas of the brain, expressly the hippocampus—an important region for learning and memory—represent the direct cause of clinical manifestations in AD [16,23,28].

Accumulating Aβ lesions are involved in complex interactions that greatly affect the neurochemical environment and the morphology of the surrounding brain tissues, resulting in the cognitive and behavioral features of AD [81] (Figure 3). Synaptic dysfunction in AD originates from the high sensitivity of axons and presynaptic oligomers, particularly long-range glutamatergic projecting axons of pyramidal neurons, to insoluble and soluble oligomeric species of Aβ. Both Aβ plaques and soluble Aβ oligomers induce synaptic swelling and dystrophic changes, making these sites favorable for the ectopic release of glutamate, which aggregates in the extracellular space [68,81]. On the other hand, presenilin 1 (PS1)—the catalytic component of γ-secretase, the enzyme responsible for the final processing of APP to produce Aβ—directly interacts with ryanodine receptors and synaptic vesicle release machinery protein (synaptotagmin 1) in neurons to modulate the release of neurotransmitters at the synapse. PS1 also interacts with the major glutamate transporter in the central nervous system (CNS), glutamate transporter 1 (GLT-1), also known as excitatory amino acid transporter (EAAT), at the PS1 large cytosolic loop located between the 6th and 7th transmembrane domains [36]. This interaction occurs both in neurons and astrocytes, resulting in altered expression and function of GLT-1, which drastically affects brain metabolism and synaptic signaling in a fashion that contributes to excitotoxicity and neurodegeneration [36,81]. In particular, downregulation of GLT-1 entails enhancement of glutamate signaling through the activation of NMDA receptors. The latter is a principal element of the memory formation system in the brain that functions through glutamate-mediated neurotransmission. Glutamate is a major excitatory neurotransmitter, and its transport from the extracellular space into the cytoplasm induces neuronal hyperactivity at the initial stages of AD pathogenesis [36,81,82].

Aβ-related alterations of glutamate signaling limit synaptic plasticity and stimulate pathological synaptic transmission through disruption of calcium homeostasis [81]. Calcium (Ca^2+^) is a second messenger that plays a core role in the regulation of various basic neuronal processes, e.g., cellular differentiation, proliferation, growth, survival, apoptosis, gene transcription, synaptic plasticity, and membrane excitability [83]. The homeostasis of extracellular free Ca^2+^, as well as axon and dendrite development, are regulated by Ca^2+^-sensing receptor (CaSR) [84]. Neurons are highly sensitive to any change in Ca^2+^ levels, and even minor changes can destructively alter neuronal activity [83]. Amyloid pathology in AD contributes to the dysregulation of neuronal Ca^2+^ signaling [81,85]. On the other side, the binding of both exogenous and accumulating Aβ42 oligomers to CaSR in neurons and astrocytes at the plasmalemma activates a set of intracellular signaling cascades such as iNOS, vascular endothelial growth factor-A (VEGF-A), and GTP cyclohydrolase 1. As a result, many cytotoxic effects ensue, including increased production of NO, increased vascular permeability, aggravated intracellular influx and aggregation of Aβ, and diminution of Aβ proteolysis [84]. Moreover, distorted Ca^2+^ channels contribute to neuroinflammation and neurodegeneration by causing neuronal membrane disruption, which allows irregular transfer of Ca^2+^ to occur and alters the release of neurotransmitters, ending with the activation of apoptotic signaling cascades [81,85]. Polymerization of the dimers of CaSR occurs at the ER [84], whereas synaptic loss in AD embraces increased Ca^2+^ levels in the ER and decreased neuronal store-operated Ca^2+^ entry [83].

On one hand, Aβ activates the phosphorylation of MAP-2, a protein that is abundant in neuronal axon terminals, leading to axon collapse and dendrite instability. Aβ oligomers induce local production of cytokines and free radicals, which directly cause synaptic swelling. PS1, the catalytic component of γ-secretase, modulates the release of neurotransmitters by directly interacting with ryanodine receptors and synaptotagmin 1 in neurons. PS1 also stimulates the ectopic release of glutamate by interacting with GLT-1, the major glutamate transporter in the central nervous system. On the other hand, Aβ pathology triggers locational and functional alterations in nAChRs by causing perturbations of the cholesterol and phospholipid components of the biomembrane, where nAChRs are located, which result in oectopic glutamate release and Ca^2+^ influx. These events lead to synaptic destruction and impair signal transduction in a manner that promotes ROS production, tauopathy, amyloidogenesis, and apoptosis.

Cholinergic transmission is modulated by acetylcholine (ACh). ACh plays a major role in cortical activation and arousal, which are involved in higher brain functions such as memory, learning, and attention, as well as in overall brain homeostasis and plasticity [86]. The balance of ACh is regulated by its synthesis through the activities of choline acetyltransferase (ChAT) and degradation by acetylcholinesterase (AChE) [87]. Nicotinic acetylcholine receptors (nAChRs) are located in the lipid-ordered domains of the biomembrane, where Aβ accumulation takes place. Aβ pathology causes perturbations of the cholesterol and phospholipid components of the membrane, leading to locational and functional alterations of nAChRs [68]. In the same way, degeneration associated with NFTs causes progressive death and dysfunction of cholinergic neurons in the forebrain and neocortex, leading to a widespread presynaptic cholinergic denervation [86]. Therefore, the characteristic cognitive dysfunction in AD is mostly attributed to dysregulation of the cortical cholinergic system caused by the deficiency of its main neurotransmitter [88]. A group of the few approved medications of AD is based on increasing ACh levels [86]. ACh has a neuroprotective effect against Aβ toxicity, both in cholinergic and noncholinergic cells, by enhancing the soluble Aβ peptide conformation and discouraging the aggregation-prone β-sheet conformation [89]. The homomeric α7- nAChR is the most widely used form of nAChRs, and has high permeability to Ca^2+^ and high affinity for Aβ. Elevated astrocytic α7-nAChR in the hippocampus, entorhinal cortex, and temporal cortex of sporadic AD patients and carriers of the Swedish APP 670/671 mutation suggest its involvement in AD pathology. Nonetheless, evidence denotes that it could be either neuroprotective or neurodegenerative, depending on Aβ levels, which vary according to disease stage. Experimentally, activation of α7-nAChRs by picomolar levels of Aβ42 promotes long-term potentiation and memory via spontaneous intracellular Ca^2+^ and NMDA signaling. On the other hand, inhibition of α7-nAChRs occurs by low micromolar levels of Aβ42 and Aβ25-35, which is followed by intercellular Ca^2+^ influx and ectopic release of glutamate, resulting in the destruction of synaptic spines out of accelerated production of NO, p-tau oligomers, and the activity of caspase-3 [84], a protein that belongs to the cysteine-aspartic acid protease (caspase) family, which, when activated, interacts with caspase-8 and caspase-9 to contribute to the execution-phase of cell apoptosis [28].

## 4. Current Treatments of AD

AD is not currently curable [15,90], and no new AD drugs have been approved in the last 16 years. Meanwhile, disease-modifying treatments, which aim to prevent neurodegenerative processes at early stages before the development of clinical manifestations, are lacking [91]. The available pharmacotherapy consists of two classes: cholinesterase inhibitors (donepezil, galantamine, rivastigmine, tacrine) that increase brain levels of ACh, and glutamatergic antagonist (memantine) that protects neurons against glutamate mediated excitotoxicity. Nonetheless, these drugs fail to treat the underlying causes of AD and only provide a symptomatic relief [18,26].

Researchers have desperately attempted to develop alternative pharmacologic treatments of AD. However, failure rates of drug development programs are exceptionally high. A current review that examined clinicaltrials.gov for all pharmacologic AD trials in 2019 revealed that 132 agents are being tested in trials that mostly include preclinical and prodromal AD. Out of the tested compounds, 19 agents target cognitive enhancement, 14 target neuropsychiatric and behavioral symptoms associated with AD, 38 agents target Aβ, and 17 agents target tau; agents targeting Aβ and tau include small molecules and monoclonal antibodies or biological therapies [91]. In other words, not a single drug among the available AD treatments is able to address the disease in its active phase or from its different pathological dimensions. It is worth noting that treatments that enhance brain Aβ clearance activity (e.g., Aβ immunotherapy) have serious adverse effects. For instance, therapeutic antibodies may cause vasogenic edema, microhemorrhage, and neuronal hyperactivity [28]. Therefore, attention has been recently directed toward natural resources to treat the underlying causes of AD and prevent its development in vulnerable people [18]. Research has recently shown that dietary intervention may promote cognitive health and prevent AD [92].

## 5. Royal Jelly, Its Ingredients and Pharmacological Properties

Royal jelly (RJ) is a white or yellowish gelatinous substance secreted from the mandibular and hypopharyngeal glands of young nurse worker bees (*Apis mellifera*). It has a pungent smell, a distinct sweet-sour taste, and an acidic pH (3.4–4.5) [93,94]. RJ possesses a wide range of health-promoting activities: antioxidant, anti-inflammatory, neurotrophic, hypotensive, antidiabetic, antihypercholesterolemic, antirheumatic, antitumor, antifatigue, antimicrobial, nematocidal, and anti-aging [95,96]. Consequently, RJ and its major active compounds have been used to attain therapeutic benefits in cancer, diabetes, hypertension, hyperlipidemia, skin diseases, etc. [63,97].

Water comprises the greatest part of crude RJ (50–70% *w/w*) [96,98,99]. Proteins represent the main active ingredient of RJ, accounting for 50% of its dry matter weight. Nine water-insoluble proteins constitute the majority of RJ protein content, which are known as major royal jelly proteins (MRJPs) [96]. Small amounts of other proteins exist in RJ such as royalisin, jelleines, and aspimin [97]. Novel proteins that do not belong to the MRJPs group have been newly discovered [100]. The antioxidant effect of RJ is attributed to its protein fraction—up to 29 antioxidative peptides have been identified in RJ hydrolysates. They contain a phenolic hydroxyl group, which possesses the ability to scavenge free radicals by donating a hydrogen atom [97]. The lipid fraction of RJ accounts for 3–6% and 7–18% of its wet and dry weights, respectively [99,101]. Short hydroxyl fatty acids constitute 80–85% of this fraction, while the rest consists of phenols (4–10%), waxes (5–6%), steroids (3–4%), and phospholipids (0.4–0.8%) [102]. Trans-10-hydroxy-2-decenoic acid (10-HDA) is the main fatty acid in RJ; it exhibits a plethora of biological properties, e.g., anti-aging, neurogenic, anticancer, antiobesity, antibacterial, and many others [63,98]. In addition, fructose and glucose constitute 90% of the carbohydrates fraction of RJ (7.5–16%) [63,97,102]. RJ contains small amounts of vitamins, minerals, phenols, esters, aldehydes, ketones, and alcohol. Numerous bioactive substances are available in RJ such as ACh and nucleotides. Nucleotides exist either as free bases such as adenosine and guanosine, or phosphates such as adenosine diphosphate and adenosine monophosphate [63,101,102]. Additionally, RJ contains currently undefined ingredients [100]. Table 1 illustrates the structure of RJ in detail.

Various factors influence the quality of RJ by affecting its ingredients and their activity. For instance, feeding bees with sugars causes significant alterations in the amounts and structure of vital constituents of RJ such as amino acids (e.g., tryptophan and lysine), derivatives of amino acids (e.g., pyroglutamic acid), amines (e.g., cadaverine), carbohydrates, and vitamins [103]. Of importance, RJ should be stored in a frozen state in order to retain its biological properties. Storage at 5 °C or above markedly reduces its soluble nitrogen and free amino acids due to activation of enzymatic degradation and interactions between its lipid and protein fractions, eventually ending with a darker, rancid, and more viscous substance that consists of water insoluble nitrogenous compounds [94].

RJ is the only food consumed by bee queens throughout their entire life [63]. It is suggested that RJ contributes to the unique qualities of bee queens: long lifespan, high fertility, and excellent learning and memory ability [104]. Hence, this review aims to investigate the anti-aging properties of RJ with a focus on cognitive function in advanced aging and AD. In this respect, it reviews, synthesizes, and discusses the most relevant studies that examine the effect of RJ on cognitive aging and AD pathology, both in cell cultures and animal models, as well as in humans when possible. It also elaborates on the molecular changes that lie behind these effects.

## 6. Effect of Royal Jelly on Cognitive Performance and Related Biological Markers

### 6.1. Evidence from Preclinical Studies

Research denotes that queen bees, which feast on RJ throughout their entire lives, demonstrate up to 5-fold higher learning and memory abilities than worker bees, which feast on honey and pollens. The exceptional cognitive abilities of queen bees are associated with persistent expression of Dnmt3 gene encoding DNA methyltransferase-enzyme catalyzing DNA methylation, which plays a principle role in the formation of long-term memory [104]. Interestingly, supplementing the diet of bee workers with RJ significantly increased olfactory learning and memory [105,106], and accelerated the expression of learning and memory-related genes (GluRA and Nmdar1) compared with control bees. Higher concentrations of RJ (20%) produced better effects than lower ones [106]. A number of studies have utilized various animal species as models of natural aging and AD (Figure 4) in order to test the effect of RJ and its elements on cognitive function. The findings of these investigations highlight the ability of RJ to enhance learning and memory retention, as well as to prevent and treat cognitive behavioral deficits. Rodents have been widely used as models of natural aging and AD.

RJ significantly improved spatial learning and enhanced memory retention by up to 48.5% in normally aged rats [107,108,109,110]. It also corrected cognitive deficits in different AD models such as copper and cholesterol-fed rabbits [70], streptozotocin-induced hippocampal neuronal death [111], trimethyltin-induced hippocampal neuronal death [112], and double transgenic APP/PS1 mice, which express two mutations associated with early-onset AD—chimeric mouse/human APP (Mo/HuAPP695swe) and human PS1 (PS1-dE9) [28]. Cognitive enhancement/recovery manifested in the improvement of the spontaneous alternation rate in the Y-maze [112], shorter time and path lengths to find an underwater escape plate in the Morris Water Maze test, increased time spent in the target quadrant and increased time of the first entrance to the dark room one week after receiving an electrical shock on the step-down passive avoidance test (the probe trial), increased crossings by up to 177.4% [28,108,109], as well as decreased time searching for food, and improved response rate to sudden sound on open field test [95] (Table 2).

The improvement of cognitive function associated with RJ treatment results from the amelioration of oxidative damage and increase of antioxidant capacity [87,106,111]. In vivo studies involving AD models show that RJ enhanced the endogenous production of antioxidants, increasing superoxide dismutase (SOD) by up to 27% and reducing free radicals such as ROS, NOS, and malonaldehyde (MDA) in the cortex and hippocampus [70,95]. A limited number of in vitro studies examined the mechanistic effect of RJ on AD pathology. The results go in agreement with those reported by in vivo experiments. RJ protected LPS-stimulated BV-2 microglia against oxidative stress and attenuated microglial inflammatory processes that contribute to amyloidgenesis. It significantly decreased the production of ROS, NOS, IL-6, IL-1β, and TNF-α and increased the mRNA expression of antioxidants such as heme oxygenase-1 (HO-1) and glutathione peroxidase (GSH-Px) [113,114]. The major fatty acid in RJ, 10-HDA, protected the BBB against LPS-induced leakage by inhibiting the degradation of tight junction proteins (occludin, claudin-5 and ZO-1) in LPS-stimulated C57BL/6 mice, as well as by increasing the expression of tight junction proteins, and decreasing the expression of chemokines (CCL-2 and CCL-3), adhesion molecules (e.g., intercellular adhesion molecule-1 (ICAM-1) and vascular cell adhesion molecule-1 (VCAM-1)), and matrix metalloproteinases (MMP-2 and MMP-9) in LPS-stimulated human brain microvascular endothelial cells (HBMECs) [114].

RJ decreased Aβ synthesis and enhanced its clearance in AD rat models, which is associated with a reduction in the total size and number of cortical and hippocampal SPs [28,95]. In a unique study, synchronous CL2006 worms were used as a nematode AD model. Supplementing aged worms with RJ/enzyme-treated RJ (eRJ) (2 mg/mL and 1 mg/mL) for 10 days at 20 °C decreased the total amount of Aβ species by 13.61% and 21.90%, respectively. Both RJ and eRJ reduced paralysis induced by Aβ toxicity and increased levels of soluble proteins by 27.13% and 21.27%, respectively [92]. Similarly, an in vitro study showed that purified RJ peptides (RJPs) (1–9 μg/mL) can interfere with the process of Aβ formation and prevent the external production of Aβ1-40 and Aβ1-42 in N2a/APP695swe cell cultures (which produce high levels of APP in AD, because they are stably transfected with the human APP gene) via down-regulation of β-secretase [115]. Another study reported increased the clearance of soluble Aβ by a DMSO-soluble fraction of RJ [116]. In addition, RJ corrected pathologies that promote amyloid cleavage, e.g., it decreased cholesterol levels [70] and increased free thyroxine (fT4) in hypothyroidism rat models [73]. MRJPs contributed to DNA repair in aged rats by increasing levels of xanthosine, which supports the anabolism of nucleic acid [109]. RJ activated autophagy genes [117,118] and protected against neuronal apoptosis [28] (the mechanisms underlying these effects are described in detail in Section 7).

### 6.2. Evidence from Clinical Trials

Preclinical trials show that RJ/eRJ, 10-HDA, RJ peptides, and MRJPs exhibit significant effects against AD pathology in animal models and in cell lines by interfering with amyloid synthesis and protein misfolding, enhancing amyloid clearance, and correcting pathologies that contribute to amyloidogenesis, such as inflammation and oxidative stress (Table 2). However, concerns have arisen about the notion that similar effects can be obtained in humans. Unfortunately, research using RJ to treat cognitive impairment in humans is scarce. In a single randomized clinical trial (RCT), 66 patients (50-80 years old) with MCI received a daily capsule of Memo^®^, a triple combination of 750 mg of lyophilized RJ with standardized extracts of *Ginkgo biloba* (120 mg) and *Panax ginseng* (150 mg), for 4 weeks. Memo^®^ significantly improved scores of the Mini-Mental State Examination compared with placebo treatment [125]. Similarly, daily consumption of 2 capsules of Lady 4—a combination of 200 mg of lyophilized RJ, 250 mg of evening primrose oil, 100 mg of *Turnera diffusa*, and 50 mg of *Panax ginseng*—for 4 weeks significantly reduced total scores of the Menopause Rating Scale. This scale measures menopausal symptoms including impaired memory, poor concentration, nervousness, depression, and insomnia. However, the authors only reported the total score of the scale, and it is unclear if Lady 4 had any effect on cognitive and psychological symptoms of menopause [127]. Another uncontrolled, open-label trial treated 55 postmenopausal women (having menopausal complaints) with Melbrosia, a dietary supplement that combines RJ with flower pollen and fermented flower pollen, for 3 months. Melbrosia significantly improved participants’ scores on the problem-solving subscale of the Frankfurt Self-concept Scale, relieved depressive and menopausal symptoms, decreased total cholesterol and low-density lipoproteins, and increased high-density lipoproteins and triglycerides, whereas VCAM-1 and C-reactive protein levels were not affected [126]. In contrast, a former RCT reported no effect of Melbrosia on biochemical parameters in women with severe menopausal symptoms, although it significantly improved vitality and relieved symptoms of headache, urinary incontinence, and vaginal dryness [128]. Minor adverse effects were reported including weight gain (a Melbrosia tablet contains 307 kcal), brief GI discomfort, transient facial flush, mild nausea, and mild transient headache in 6 subjects [125,126]. Nonetheless, it is impossible to confirm either the effectiveness of RJ in AD patients or its side effects from the data reported in these studies. In all trials, RJ was combined with other elements, biomarkers were not assessed in patients with cognitive impairment, and various methodological flaws were incorporated, e.g., small sample sizes, lack of blinding, and absence of a control group [125,126].

## 7. Mechanisms Underlying Effects of RJ on Cognition and AD-Related Pathology

The findings from the studies examined in this review indicate that RJ targets a variety of pathophysiological mechanisms of cognitive aging and AD. The neurotrophic, antioxidative, anti-inflammatory, anti-apoptotic, and anti-amyloidogenic properties of RJ allow it to act as a multidomain cognitive enhancer, which may delay the onset of AD, slow its progression, and foster recovery. Figure 5 summarizes the anti-AD and cognitive promoting activities of RJ. This section describes the mechanisms underlying these activities in detail.

RJ improves cognition via a network of inter-related mechanisms. It alleviates Aβ pathology by (1) decreasing its influx through the BBB via inhibition of RAGE, (2) preventing the cleavage of APP into Aβ via inhibition of BACE1, and (3) facilitating the degradation and clearance of Aβ by IDE, NEP, and LRP1. From another perspective, RJ activates AMPK, a master signaling pathway that activates various signaling pathways (mainly through an indirect pathway mediated by phosphorylation of FOXO, which inhibits mTOR). AMPK activity promotes autophagy and antioxidant production, and suppresses microglial inflammation via inhibition of various oxidative, inflammatory, and apoptotic pathways, e.g., iNOS and NF-κB. RJ and its lipids bind to estrogen receptors β and α to enhance the production of neurotrophins such as NGF and BDNF, which promote ACh production, neurogenesis, and synaptogenesis, and counteract amyloidogenesis, etc.

### 7.1. Royal Jelly-Related Neuroprotection Is Mediated by Regulating the Production of Neurotrophins

The depletion of neurotrophins such as brain derived neurotrophic factor (BDNF), glial cell line-derived neurotrophic factor (GDNF), and nerve growth factor (NGF) significantly affects neuronal and nonneuronal responses to AD and accelerates disease progression [9,15]. On the other hand, the enhancement of the production of endogenous neurotrophins limits neurodegeneration. NGF, a key member of the neurotrophin family, promotes the survival and function of cholinergic neurons of the basal forebrain, which undergo massive degeneration in AD [129]. NGF boosts the production of ACh in cholinergic neurons by enhancing the release and activity of ChAT [85]. In fact, RJ (which stimulates NGF production [130,131]) was shown to increase the activities of ChAT and decrease AChE in the cortex of a rabbit model of AD induced by ovariectomy and cholesterol diet [95]. NGF also enhances healing of injured astrocytes in different brain areas [132]. The highest levels of NGF originate from brain regions that are most insulted in AD: the cortex, hippocampus, and the basal ganglia [133,134]. External NGF replacement has been used as a potential treatment to promote neural regeneration in neurodegenerative disease such as AD [133,134]. However, the passage of NGF across the BBB represents a major challenge for nonviral gene delivery via a systematic transvascular route. Therefore, the delivery of this therapy is limited to invasive intracerebral injection [135]. Nonetheless, its efficiency is also questionable. For example, a two-year RCT examined the effect of intracerebral injections of adeno-associated viral vector (serotype 2)-NGF in AD patients using magnetic resonance imaging, fludeoxyglucose F18-labeled positron emission tomography imaging, and neuropsychological testing. The results revealed that NGF was safe and well-tolerated, but there was no evidence of efficacy [133]. Therefore, the use of natural agents that are capable of stimulating internal production of NGF may support healing of injured brain tissues in a more cost-effective manner than external NGF administration.

AMP N1-oxide is a chief RJ derivative that can modulate neuronal function and stimulate nitrite outgrowth and processes formation via adenosine A2A receptors [130,131]. Targeting glial A2A receptors, in particular, is a promising target for AD treatment. A2A receptors are expressed in the frontal cortex and hippocampus, and they play a major role in neuroinflammation and neurodegeneration by controlling synaptic plasticity, NMDAR activity, and glutamate uptake by astrocytes [84]. Strong activation of A2A receptor genes by eRJ and AMP N1-oxide highlights their role in the regulation of synaptic plasticity, which is crucial for early neuronal development [130,136]. The effects demonstrated by AMP N1-oxide at concentrations of 20 or 40 mM are similar to the effect of 10 or 50 ng/mL of NGF. Meanwhile, combining NGF and eRJ had no effect on the percentage of process-bearing cells compared with solo NGF treatment, but a marked increase in the percentage of cells with processes longer than two times the cell diameter was recorded [131]. Evidence denotes that AMP N1-oxide mimics the regenerative effect of NGF [130]. NGF stimulates neurite outgrowth in neuronal progenitor stem cells (PC12) and contributes to synaptic plasticity through binding receptors p75 and tropomyosine-related kinase A (TrkA), a member of the tyrosine kinases family [130,137]. This process entails increasing the expression of a protein of mature neurons known as neurofilament M and enhancing the differentiation of pheochromocytoma PC12 into neurons similar to sympathetic neurons. Effects of AMP N1-oxide on cell outgrowth and differentiation involve the activation of two main cellular signaling cascades: adenosine A2A receptor-mediated phosphatidylinositol 3-kinase (PI3K)/Akt/cAMP-dependent protein kinase (PKA)/CREB and mitogen-activated protein kinase (MAPK)/extracellular signal-regulated kinase 1 or 2 (ERK1/2) signaling pathways [130]. Phosphorylation of ERK1/2 or the p38 MAPK pathway for CREB takes place via an indirect pathway mediated by the phosphorylation of pp90 ribosomal S6 kinase (p90RSK), mitogen- and stress-activated protein kinase (MSK)1/2, MAPKAP kinase 2 [138]. Integrin receptor signaling is necessary for the activation of ERK1/2, because it stimulates Mn^2+^ to accelerate neurite outgrowth from PC12 cells treated with eRJ or AMP N1-oxide [136]. In addition, AMP N1-related activation of PI3K/Akt upregulates phosphorylation of glycogen synthase kinase-3β (GSK-3β) to enhance cellular survival by abrogating inflammatory signaling [139].

BDNF, especially in the hippocampus, plays a major role in the regulation of processes involved in learning and memory, such as long-term potentiation, synaptic plasticity, axonal sprouting, and proliferation of dendritic arbor, mainly through its interaction with Trk B receptor [9]. Polymorphisms of *BDNF* have been reported to be associated with AD [140,141]. RJ, 10-HDA, and 10-HDA-related esters (e.g., Trans-2-decenoic acid ethyl ester (DAEE) and 4-Hydroperoxy-2- decenoic acid ethyl ester (HPO-DAEE)) are reported to generate intracellular signals like BDNF, which grants them the potential of neuronal regeneration. In addition to stimulating neurogenesis and neurite outgrowth, these compounds promote neuronal survival and demonstrate neuroprotection against physical injury and neurotoxicity [112,136,142,143]. A comparison of signaling induced by BDNF and DAEE in cultured rat embryonic neurons revealed that DAEE increased the mRNA expression of BDNF and neurotrophin-3 and enhanced synaptogenesis as portrayed by increased synapse-specific proteins such as synaptophysin, synapsin-1, and syntaxin [144]. Evolving evidence indicates that RJ-related induction of BDNF signaling may alleviate AD symptoms in APP/PS1 mice by reducing cortical and hippocampal levels of BACE1, soluble and insoluble Aβ40 and Aβ42, as well as the number and size of Aβ plaques [28].

GDNF is a diffusible peptide, which is centrally involved in neuronal differentiation and survival; it has been suggested that it regulates processes implicated in the repair of damaged brain tissues [145]. Research shows that oral administration of RJ enhances the production of GDNF and neurofilament H in the hippocampus of adult mice [146]. Interestingly, the expression of neurofilament M was associated with increased differentiation of neural stem/progenitor cells (NS/NPCs) into mature neurons [130]. NS/NPCs exist in the adult hippocampal dentate gyrus (DG), and can differentiate into neurons that regulate memory and learning [112]. Injecting umbilical cord blood mononuclear cells (UCBMCs) transduced with adenoviral vectors expressing GDNF to the sites of neurodegeneration into AD transgenic mice resulted in the presence of UCBMCs in the hippocampus and cortex several weeks after transplantation. This treatment induced long-lasting neuroprotection and stimulated synaptogenesis, delineated by the restoration of postsynaptic density protein 95 and synaptophysin levels in the hippocampus, which was associated with improvement of spatial memory [145].

### 7.2. Royal Jelly Regulates Neurotransmission in Models of Advanced Aging and Alzheimer’s Disease

Gamma-aminobutyric acid (GABA) is a major brain neurotransmitter that is synthesized inside neurons by glutamate decarboxylase (GAD) and metabolized by GABA-transaminase (GABA-T) [122]. Current research highlights the role of GABA in memory and spatial learning. RJ treatment increased GABA levels in the cortex of rats undergoing cortical damage induced by tartrazine [120]. In contrast, RJ supplementation to naturally aged rats decreased GABA in the striatum and hypothalamus. This effect was not associated with a drop in glutamate, the precursor of GABA [122]. It seems that minor changes in GABA neurotransmission affect glutamate-mediated neurotransmission via NMDAR activity [82]. Different GABA-ergic receptors work in a fashion that sustains the balance between the excitatory glutamatergic and the inhibitory GABAergic neurotransmitters [122]. In this regard, antagonism of GABA A receptor by injecting bicuculline into CA1 altered spatial memory and maintained nonspatial memory under conditions of NMDAR antagonism by ketamine. GABA A receptor agonism, under the same conditions, did not alter nonspatial change detection ability but altered spatial novelty [82]. In another study, agonism of GABA A and GABA B receptors by intra-CA1 microinjections of muscimol and baclofen impaired memory retention in the same way as D-AP5 (a NMDAR antagonist). Meanwhile, agonism of GABA A receptor accelerated the memory impairment effect of D-AP5, whereas its antagonism had no effect. The activation and blockade of GABA B receptor by baclofen and phaclofen at a low dose (0.1 µg/rat) potentiated memory impairment induced by simultaneous administration of low doses of D-AP5 (0.0625 and 0.125 µg/rat); however, they abolished the memory impairment effect of higher D-AP5 doses (0.25 µg/rat). Thus, the agonism or antagonism of different GABAergic receptors may affect memory in a dual manner, depending on the level of NMDAR neurotransmission [147]. It is possible that RJ modulation of GABA in the hypothalamus is associated with increased activity of the hypothalamus and enhanced production of gonadotropin-releasing hormone and sex steroids, which express memory promoting activities (detailed in Section 7.8.) [122].

The destruction of synaptic spines in AD brains results from increased activity of free radicals, phosphorylated tau oligomers, and caspase-3, which follow excessive intercellular Ca^2+^ influx. Synaptic loss triggers dysregulation of synaptic signal transduction, i.e., the synthesis, transport, release, and uptake of neurotransmitters, which accelerates neurodegeneration [73,80,84]. Evidence shows that serotonergic transmission, which plays a role in the pathogenesis of depression [148], influences processes of learning and memory. Serotonin receptors exist in the prefrontal cortex, amygdala, and hippocampus—the main brain structures involved in high cognitive functions [121,149]. Accumulating evidence suggests that agonism of 5-hydroxytryptamine (5-HT)2A/2C and 5-HT4 receptors or antagonism of 5-HT1A, 5HT3, and 5-HT1B receptors protects against memory impairment and promotes learning under conditions of high cognitive demand [149]. Meanwhile, 90% of the metabolism of tryptophan, the precursor of serotonin, is directed toward the synthesis of kynurenine. Kynurenine is a precursor of kynurenic acid, which affects cognition by antagonizing glutamate ionotropic receptors. In this light, tryptophan deficiency is associated with increased memory impairment [149]. Glial activation and high levels of cytokines contribute to the upregulation of astrocytic serotonin transporter (5-HTT) and the reduction of extracellular serotonin/5-HT levels [148]. From another standpoint, evolving knowledge highlights an important role of dopamine and its D2-like dopamine receptor gene (AmDOP3) in learning and memory via the homovanillyl alcohol target dopamine pathways in the brain [150]. RJ contains tryptophan and tyrosine, the precursors of serotonin and dopamine [63,103,151]. Treating experimental models with RJ and tyrosine significantly increases brain levels of dopamine and its metabolites [152,153]. Therefore, it is possible that enhancing brain neurotransmission is one of the molecular changes that lie behind the cognition-agonizing effects of RJ. In two studies, both short- and long-term application of RJ to naturally aged rats significantly improved learning, spatial memory, and motor performance. Such effects were associated with significant modifications in brain content of various neurotransmitters and their metabolites; the serotonergic and dopaminergic activity increased in the prefrontal cortex, as denoted by a decrease of 5-HT and dopamine and an increase of their metabolites—5-hydroxyindoleacetic acid (5HIAA) and 3,4-dihydroxyphenylacetic acid (DOPAC)—and increased turnover of these metabolites. RJ also decreased cortical levels of 3-methoxy-4-hydroxyphenylglycol, a metabolite of noradrenaline, and its turnover. Striatal DOPAC increased, which is in line with the improvement of motor activity by RJ treatment. Short-term RJ treatment increased hippocampal 5-HT content and decreased serotonin turnover but had no effect on levels of dopamine or noradrenaline though their metabolism in the prefrontal cortex, hippocampus, and striatum changed [107,121]. In another study, improvements in learning and memory following RJ treatment were associated restoration of brain levels of noradrenaline and dopamine in a mouse model of aging induced by intraperitoneal injection of d-galactose [124]. In two other studies, RJ restored cortical levels of serotonin, dopamine, and noradrenaline and attenuated cortical neuron death induced by tartrazine and cadmium [119,120]. Overall, these reports suggest that RJ treatment of cognitively impaired animals was associated with less synaptic destruction and enhancement of synaptogenesis—a finding reported in several previous studies [130,136,144,154].

### 7.3. Royal Jelly Regulates Energy Metabolism in the Brain

Glucose is necessary for neuronal functioning, and glucose transporters (mainly GLUT1, and to a limited extent, GLUT2, GLUT 3, GLUT4, GLUT8) mediate glucose transport across the BBB and glucose uptake into neurons and astrocytes [64,155,156]. Higher metabolic rates and higher predisposition to deposit fat along with changes in the allocation of energy supplies are essential for the evolution of brain size and complexity [157]. Human carriers of APOE-ε4 experience decreased glucose metabolism in the temporal cortex and parahippocampal gyrus early, i.e., during young adulthood and middle-age [21]. Meanwhile, glucose and oxygen metabolic rates in brain cells decrease with normal aging and markedly diminish in AD [157,158]. On one hand, aging entails increased expression of RAGE, which affects various major metabolic functions (e.g., glucose tolerance) in conditions predisposing to AD such as obesity, metabolic syndrome, and type 2 diabetes [64,159]. In fact, diabetes mellitus is associated with alterations in the brain expression levels of different glucose transporters, which contribute to poor glucose supply to the brain [64,155]. On the other hand, AD embraces deficiency of adiponectin, an adipose tissue-derived adipokine that plays major roles in inflammation, oxidative stress, lipid and glucose metabolism, and neurogenesis. Moreover, it activates peroxisome proliferator-activated receptor-α (PPAR-α) signaling in microglia exposed to toxic Aβ oligomers. Reduced levels of adiponectin, such as in obesity and type 2 diabetes, worsen cognitive impairments through inhibition of AMP-activated protein kinase (AMPK), which is associated with the development of cerebral insulin resistance [33].

In light of these reports, processes of glycolysis and gluconeogenesis evidently affect higher brain functions, e.g., learning and memory [109]. Evidence has revealed that a slight increase of brain glucose supply, indicated by elevated energy consumption in neurons of the mushroom body (i.e., the main memory center in fruit flies), is necessary for long-term memory formation. This effect is mediated by dopamine signaling [160]. RJ reduced amyloidogenesis in aged *Caenorhabditis* (*C.*) *elegans* by inhibiting IIS signaling [92]. Downregulation of IIS improves insulin sensitivity [63] and triggers local transfer of Ca^2+^ from the ER to mitochondria to stimulate GLUT4 translocation to the cell surface to increase glucose uptake [161]. Supplementing aged rats with MRJPs protected against memory impairment through restoration of brain levels of adenosine triphosphate (ATP) and improvement of brain metabolism. Mechanistically, MRJPs modestly enhanced glucose supply to the brain, resulting in enhanced brain levels of phosphoenolpyruvic acid, which stimulates gluconeogenesis to modulate its level. Metabolomics analysis revealed that urine metabolites of MRJPs-treated rats were similar to those of young rats, which significantly differed from the untreated control aged rats [109]. Experimental trials showed that RJ can support neuronal metabolic activities in AD brains. In an AD rabbit model induced by copper and a high cholesterol diet, *N*-acetyl aspartate and glutamate markedly dropped and choline and myo-inositol increased compared with the normal control group. On the other hand, RJ treatment significantly improved neuronal metabolic activities, as shown by increased levels of *N*-acetyl aspartate and glutamate and their ratio to creatine, whereas levels of choline and myo-inositol and their ratio to creatine decreased compared with untreated AD rabbits [70]. Furthermore, cumulative knowledge denotes the contribution of royal jelly to the production of ketone bodies [162]. AD brains utilize ketones, such as acetoacetate and β-hydroxybutyrate, as major alternative energy substrates to glucose [158]. Empirical evidence shows that consumption of a ketogenic diet improves verbal memory and processing speed in patients with cognitive impairments [163,164].

One of the mechanisms through which RJ controls energy expenditure involves the modulation of eat-2 mutant, which activates multiple dietary restriction-related signaling cascades [63,165]. Brain cells adaptively respond to bioenergetic challenges such as stress induced by food deprivation through activation of signaling pathways that enhance synaptic structure and function, stimulate synaptic formation, increase the production of new neurons from stem cells, and promote neuronal resistance to metabolic, oxidative, excitotoxic, and proteotoxic stresses which are in the pathogenesis of disorders such as AD [157]. Experimentally, dietary restriction in aged rats was shown to increase cortical expression of GLUT1, GLUT3, and GLUT4, while intermittent feeding increased AMPK phosphorylation in the hippocampus [156]. It seems that RJ contributes to the regulation of neurons under conditions of low energy supply. In this context, positive effects of RJ (e.g., suppression of inflammation and oxidative stress, increased autophagy, etc.) have been associated with the activation of AMPK [114]. AMPK is an energy sensing pathway that gets activated under conditions that entail deficiency of cellular nutrients. It is a downstream effector of IIS [63]. AMPK phosphorylates Forkhead Box O transcription factor (FOXO) to induce a catabolic response for energy supply in cells low in nutrients [63,166].

### 7.4. Royal Jelly Protects Against Neuroinflammation

Accumulating evidence shows that the microglia of senescent brains are most affected by age-related activation of inflammatory signaling. High expression of microglial inflammation markers leads to dystrophic morphology (e.g., deramification, cytoplasmic fragmentation, and shortening of cellular processes). Such manifestation of microglial senescence alters their ability to promote proper neuronal function or protect neurons against the accumulation of NFT and Aβ oligomers [7,167]. Neuroinflammation broadens neuron loss by accelerating oxidative stress. A number of investigations included in this article challenged microglia or rodent models with LPS to induce a neuroinflammatory phenotype that was parallel to that in AD [113,114,117]. LPS induces inflammation by regulating the activity of NF-κB, MAPK, signal transducer and activator of transcription 1 (STAT1), and activator protein (AP-1), and c-Jun NH2-terminal kinases (JNK) [113]. Furthermore, markers of neuroinflammation are evidently high in AD models produced by toxic treatments [113,114,117,119]. In these experiments, RJ demonstrated anti-inflammatory effects by inhibiting the expression of proinflammatory cytokines (IL-6, IL-1β, and TNF-α) [113,114,117,119], chemokines (CCL-2, CCL-3) [114], and related genes of iNOS and cyclooxygenase-2 (COX-2) [113]. NF-κB and cytokines stimulate the slong-term expression of adhesion molecules such as ICAM-1 and VCAM-1 in the endothelium. ICAM-1 and VCAM-1 bind actin cytoskeleton to activate several intracellular signaling pathways that influence immunological synapse formation and cellular immune responses such as cytokine production and microvascular permeability. The latter facilitates the transfer of solutes (e.g., Aβ) and leukocytes into peripheral tissues [168,169]. Downregulation of inflammatory cytokines by RJ was associated with inhibition of the gene expression of ICAM-1, VCAM-1, MMP-2, and MMP-9, as well as with less degradation of tight junction proteins and an increase of their mRNA expression. These effects resulted in less BBB permeability [114].

RJ demonstrates an immunomodulatory effect, and its glycoproteins are reported to contain a T-antigen unit [170]. However, the anti-inflammatory effects of RJ in AD-related models are complex. On one hand, RJ suppressed the phosphorylation of some of the main inflammatory pathways such as NF-κB, p38, and JNK [28,113,117]. In particular, 10-HDA has the ability to positively affect TLR signaling by inhibiting the gene expression of TNF receptor-associated factor 1 (TRAF1) in LPS-stimulated microglia [117]. TRAF1 activates NF-κB and JNK signaling, and it negatively regulates TLR and Nod-like receptor signaling. It produces these actions by interacting with TNFR2 to form a heterodimeric signaling complex. This complex recruits cIAPs, which possess E3 ligase activity that promotes the addition of K63-linked polyubiquitin (K63-Ub) to receptor-interacting serine/threonine-protein kinase 1 (RIPK1). K63-Ub activates Inhibitor of kappa B kinase (IKK) and MAPK by recruiting transforming growth factor-β (TGF-β)-associated kinase (TAK1), TAK binding protein (TAB), and the linear ubiquitin assembly complex (LUBAC). LUBAC modifies K63-Ub to hybrid molecules and catalyzes the addition of polyubiquitin polymerized through the M1 position (M1-Ub) [171]. M1-Ub facilitates the phosphorylation of Inhibitor of κB (IκB) through the activity of IKK, which modifies it to K48-Ub. Degradation of K48-Ub stimulates the nuclear translocation of NF-κB [117,171]. Western blotting shows that 10-HDA inhibits the nuclear translocation of NF-κB p65 and NF-κB p50 subunits in activated microglia by impeding the phosphorylation and degradation of IκBα [117]. On the other hand, 10-HDA can suppress the activity of NLRP3 inflammasome-IL-1β signaling—a cytosolic protein oligomer that activates inflammatory responses by promoting the proteolytic cleavage, maturation, and secretion of inflammatory cytokines [117].

Autophagosomes initiate innate and adaptive immune responses by supplying major histocompatibility complex-loading compartments and endosomal pattern recognition receptors with intracellular pathogen-associated molecular patterns [172]. Research reports impaired autophagic influx under inflammatory conditions in AD mice [173]; 10-HDA was reported to promote autophagic immune regulation and evoke autophagy in activated microglia and LPS-treated mice by upregulating both Unc-51-like autophagy activating kinase (ULK) and microtubule-associated protein 1 light chain 3-II (LC3-II) and inhibiting sequestosome 1 (SQSTM1/p62). ULK and LC3-II are proteins that reflect activation of autophagy [117]. The former gets activated through the inhibition of mammalian target of rapamycin (mTOR) under nutrient-starving contexts [174]. SQSTM1 is a key autophagy gene that regulates the activity of many signaling cascades including cytokine signaling. It directs ubiquinated cargoes to autophagosomes for degradation. Though clearing dysfunctional proteins and organelles is important for cellular function, the removal of basic cellular protein structures represents a prodeath factor. Research shows that reduction of SQSTM1 delays brain damage induced by injury [175]. In total, these reports indicate that RJ and its constituents promote qualified cellular housekeeping under inflammatory conditions.

Evidence shows that the anti-inflammatory effects of RJ originate from compensatory changes in energy sensing pathways. For example, 10-HDA treatment of LPS-stimulated mice and HBMECs alleviated inflammation through activation of the energy sensing AMPK/PI3K/AKT signaling pathways [114]. Inactivation of AMPK in AD is associated with the severity of cognitive impairment [33]. Meanwhile, activation of AMPK improves immunometabolism and the function of immune cells. AMPK inhibits the activity of NF-κB and suppresses its related pro-inflammatory responses [176] via the activation of multiple signaling pathways including FOXO [177]. Interestingly, the effect of 10-HDA on neuroinflammation and autophagy in LPS-treated mice and microglial BV-2 cells was mediated by the upregulation of FOXO1. Chemical inhibition of FOXO1 dampened the effect of 10-HDA on NF-κB and NLRP3 inflammasome-IL-1β signaling [117]. FOXO downregulates AKT (a substrate of mTOR) through PHA-4/FOXA transcription factor and S6K, which stimulate the expression of various autophagy genes and the translation of specific mRNAs that face inflammation and preserve homeostasis by facilitating the repair or degradation of endogenous proteotoxic stress that occurs with aging [178,179,180,181].

### 7.5. Royal Jelly Protects Against Oxidative Stress

Reduced levels of endogenous antioxidants contribute to memory alteration [87]. Likewise, increased production of free radicals and oxidative stress destroy biological infrastructures (e.g., lipids, proteins, and DNA) in the CNS of AD patients [28]. Most studies in this review reported high ROS/NOS production in AD models (Table 2). A recent study treated *Drosophila Canton-S* with H_2_O_2_ or paraquat (*N*,*N*′-dimethyl-4,4′-bipyridinium dichloride, a toxic substance widely used as a herbicide) to induce an oxidative stress-model of aging similar to AD [96]. Studies involving AD models show that RJ can suppress oxidative stress as indicated by a reduction of oxidative biomarkers such as MDA, ROS, and NOS, both in the plasma and the brain, particularly in the hippocampus and the cortex [28,70,95,96,113,114,119,120,124]. RJ can also enhance the internal antioxidant capacity by stimulating the production of antioxidants such as HO-1, GSH-Px, SOD, and catalase [70,95,96,113,119,120,124]. The antioxidant activity of RJ is a principal contributor to the reported amelioration of apoptosis of cortical and hippocampal neurons in AD models [28,70,119,120]. It has been suggested that ACh and the enzyme ingredients (e.g., lipase and SOD) of RJ ameliorate oxidative damage and improve memory [106]. In addition, existing knowledge emphasizes the antioxidant properties of amino acids, peptides, and proteins in RJ [63].

It appears that RJ produces its antioxidant and anti-inflammatory effects by influencing an integrated network of different signaling pathways, which have positive feedback effects on the activity of each other. In this context, 10-HDA treatment of LPS-stimulated mice and HBMECs alleviated inflammation and resulted in numerous positive effects, including reduction of ROS emission through activation of AMPK/PI3K/AKT signaling [114]. The activation of AMPK regulates the function of nuclear factor-erythroid 2-related factor 2 (NRF2), the principal antioxidant pathway [177]. NRF2 stimulates the expression of antioxidant genes such as HO-1 and SOD [154]. In fact, RJ treatment of experimental AD models resulted in increased activity of NRF2 [119]. In addition to targeting the regulation of oxidative stress, NRF2 and related antioxidants (e.g., HO-1) directly silenced neuroinflammation by inhibiting the transcription of cytokines such as IL-6 and IL-1β [154,182]. From another aspect, RJ significantly downregulated the signaling of iNOS as a result of inhibition of inflammatory pathways [113,119]. The activation of iNOS occurs in immune activated cells, and it results in persistent emission of NO, leading to cytotoxic effects. In contrast, the activation of endothelial NO synthase (eNOS) or neuronal NO synthase (nNOS) stimulates a brief production of NO (for seconds or few minutes), which acts as a signal molecule that regulates innate immune response [183,184].

Pyridine nucleotide nicotinamide adenine dinucleotide (NAD+) is a chief molecule that contributes to health and lifespan via the regulation of normal cellular bioenergetics, scavenging of free radicals, autophagy, DNA repair, and genome stability [185]. Cognitive decline that occurs with advanced aging is associated with diminution of NAD+ and its downstream effector, calcium/calmodulin-dependent serine protein kinase in the CA1 hippocampal region [186]. Recently, supplementation of NAD+ precursors has been adopted to therapeutically increase NAD+ levels, and consequently, counteract oxidative stress [185]. Aged rats receiving MRJPs demonstrated increased levels of nicotinic acid mononucleotide (NaMN), a metabolite of nicotinic acid (NA) and a precursor of NAD+ [109]. High intracellular NA levels are cytotoxic, and its metabolism is necessary for therapeutic actions to occur [187]. It has been suggested that all anabolism of NAD+ biosynthesis goes through NaMN, which maintains intracellular redox status when catalyzed to [NAD+ (H)] by keeping a balanced NADH/NAD+ ratio [109]. The molecular mechanisms employed by NaMN to ameliorate the cytotoxic effects of oxidative stress involve the inhibition of AKT [187].

Downregulation of AKT by RJ is associated with the activation of FOXO and inhibition of the nutrient sensing mTOR pathway, a master regulator of metabolism and autophagy [63,166]. mTOR has two distinct forms: mTORC1 and mTORC2. The latter has four main proteins, namely mTOR, rapamycin-insensitive companion of target of rapamycin, mammalian stress-activated protein, and kinase interacting protein. All these proteins are located at mitochondria-associated ER membranes [166]. UPR signaling is affected by stress of the ER while experimental maneuvering of this pathway alleviates neurodegeneration in preclinical models of AD [72]. In detail, PERK homodimerizes after its activation, leading to the phosphorylation of serine residues on cytoplasmic eukaryotic initiation factor 2 alpha (eIF2α) and the phosphorylation of the activating transcription factor 4 (ATF4). ATF4 controls apoptosis by targeting CCAAT enhancer-binding (C/EBP) protein homologous protein (CHOP) and triggers adaptive programs that maintain homeostasis via regulation of redox proteins. Independent of eIF2α, PERK also activates NRF2, which modulates inflammation and oxidative stress [79]. The organized interaction of these signaling pathways further regulates the activity of AMPK to promote a cycle of activation of the basic processes involved in cellular homeostasis, such as stress resistance, antioxidative capacity, autophagy, and DNA repair [176,177].

The reduction of AGEs by RJ may be another molecular mechanism for counteracting oxidative damage. RJ treatment decreased the expression of RAGE (the main receptor for AGEs) in an AD model [70]. RAGE increases with aging, and is highly expressed in AD brains. When it binds to AGE, it activates the intracellular signaling involved in oxidative stress and inflammation. Thus, it floods the body with free radicals and inflammatory mediators like TNF-α, IL-6, and C-reactive protein, which induce microglial activation and disrupt cellular structure and Aβ metabolism [33,64]. Treating *Drosophila* intoxicated by H_2_O_2_ or paraquat with a combination of peptides of eRJ and collagen peptides from the skin of *Carcharhinus falciformis* fish reduced carbonyl proteins [96]. This finding may represent a read out of reduced activity of RAGE. RAGE binds to several other ligands such as oligomeric forms of Aβ and ligands involved in inflammatory responses, e.g., S100/calgranulins and high mobility group box 1 [64,159]. In addition, the activation of RAGE causes generalized endothelial dysfunction, which increases vascular disorders (a core AD risk factor), decreases cerebral blood flow, increases the permeability of the BBB, and facilitates the transfer of excessive amounts of Aβ from the blood to the brain [33,64]. Evidence indicates that RJ can protect against endothelial insult and restore normal endothelial function during chronic exposure to hyperglycemic conditions, which stimulate RAGE and increase the production of AGEs [33,64]. A recent study found that RJ treatment of human endothelial cells exposed to 30 mM glucose over 72 h increased the expression of LC3-II autophagy-related factor, and decreased the activity of MPP-2 and MPP-9 [118]. Altogether, these reports suggest that the inhibition of RAGE by RJ represents a multidimensional mode of cognitive enhancement: controlling oxidative stress, inflammation, glial activation, and Aβ metabolism, in addition to enhancing vascular integrity and protecting against BBB leakage under conditions of metabolic dysfunction.

### 7.6. Royal Jelly Promotes Neuronal Regeneration and Attenuates Apoptosis

The pathologies contributing to cognitive deterioration and AD development cause serious damage to the cortical and hippocampal neurons, especially in the DG [112]. The neuroprotective effects of RJ in AD models comprise improved neuronal cell structures, increased cellular survival, and reduced neuronal loss [70]. Evidence documents that RJ, AMP N1-oxide, and 10-HDA promote neurogenesis and gliogenesis by triggering the proliferation of NS/NPCs in vitro [143,188]. Experimentally, RJ enhances cognitive ability by promoting neuronal regeneration in vivo. In one experiment, mice intraperitoneally injected with trimethyltin, a toxic organotin substance that induces selective acute neuronal death in hippocampal DG, were treated with dietary RJ for 6 days. Trimethyltin decreased the number of DG neurons by 50% and caused intense cognitive dysfunction. Hippocampal DG contains NS/NPCs, which generate the neurons responsible for cognitive function. RJ treatment reversed trimethyltin-induced DG neurotoxicity by enhancing neurogenesis, resulting in functional neurons that were capable of ameliorating cognitive impairment [112]. In another study, a cholesterol diet and copper intoxication induced AD in ovariectomized rabbits, which manifested by nucleus shrinkage, neuronal loss, and the disappearance of Nissl bodies in the cortex and hippocampus. RJ treatment significantly increased the number of neurons in the hippocampal CA1, CA3, and DG regions and cortical PCL and MCL areas by 40%, 56%, 34%, 23%, and 34% respectively [70].

Immunofluorescence staining and Western blotting revealed that RJ inhibited apoptosis in the brains of AD rabbits and APP/PS1 transgenic mice; the number of activated caspase-3 immunolabeled cells, as well as the covered areas of activated caspase-3, decreased by up to 57% in the cortex and hippocampus of RJ-treated AD models compared with untreated animals [28,70]. Mechanistically, RJ amelioration of neuronal apoptosis resulted from the inhibition of the phosphorylation of IκBα, p38, and p- JNK via inhibition of the translocation of NF-κB p65 into the nucleus [28,113]. As a result, levels of the mitochondrial anti-apoptotic signaling molecule B-cell lymphoma-2 (Bcl-2) increased, and levels of pro-apoptotic signaling molecule Bcl-2-associated X protein (Bax) decreased [119]. Downregulation of the Bax/Bcl-2 ratio by RJ was associated with an evident decrease of the expression of cleaved caspase-3 in the hippocampus and cortex [28,119]. Bcl-2 promotes mitochondrial integrity, while Bax is a chief effector of the mitochondrial apoptosis pathway. Emerging knowledge signifies that AD neurons exhibit drastic ER stress, which stimulates mitochondrion Ca^2+^ fluxes and apoptotic responses by promoting Bax phosphorylation through direct conformational changes, or through the displacement of anti-apoptotic Bcl-2 proteins. Activated Bax translocates into the mitochondria to form protein-permeable channels/pores within the outer mitochondrial membrane in a process known as mitochondrial outer membrane permeabilization (MOMP). MOMP stimulates the release of pro-apoptotic factors (e.g., AIF, cyt-c, EndoG) from the mitochondrial intermembrane space into the cytoplasm to activate a caspase-dependent neuronal apoptosis in AD brains [189,190]. The investigations addressed by this review denote that the amelioration of apoptosis of cortical and hippocampal neurons in AD models is closely linked to the antioxidant and anti-inflammatory activities of RJ [28,70,119,120].

On the other hand, RJ promotes the production of neuroprotective molecules. In this regard, MRJPs increased cysteic acid in aged rats, indicating an involvement of the cysteine-taurine metabolism pathways in memory enhancement by RJ. MRJPs are rich in cystine, which can be converted into cysteine and then into cysteic acid by cysteine lyase [109]. Cysteine is used as a rate-limiting substrate for the production of the neuroprotective gaseous physiological modulator hydrogen sulfide (H_2_S) by cystathionine beta-synthase in the parenchyma of the brain [191]. In addition, cysteic acid is a precursor for taurine, a sulfur-containing, free amino acid that is abundant in excitable tissues such as the brain and the heart. Taurine deficiency is associated with multisystem failure and memory impairment [192]. Meanwhile, diets rich in taurine and cysteine enhance brain production of H_2_S [191]. Indeed, RJ treatment of ovariectomized rats increased myelin galactolipids including galactosylceramide and sulfatide—an effect that was associated with a slight increase in brain weight [87]. Taurine acts as a neuromodulator that regulates intracellular Ca^2+^ homeostasis, prevents hippocampal neuron loss, guards synaptic plasticity against excitotoxicity, decreases Aβ accumulation in the hippocampus, and enhances memory in AD experimental models through the modulation of the GABAergic system (Section 7.2) [192,193].

### 7.7. Royal Jelly Mitigates Amyloid-Related Neurotoxicity

Both in vivo and in vitro studies showed that RJ significantly decreased blood and brain levels of Aβ [28,95] and reduced the deposition of Aβ, both in advanced aging and AD models [28,92,95,115,116]. Imbalance between the formation and degradation of Aβ plays a major role in its excessive deposition in AD. RJ abrogates Aβ pathology by regulating processes essential for its production, degradation, and clearance [28]. In one study, RJ reduced levels of soluble Aβ 40 and Aβ42 by 24% and 40%, respectively. However, the efficiency of removal of insoluble Aβ 40 and Aβ42 was even much higher (60%). On one hand, RJ interferes with Aβ synthesis by decreasing the expression of BACE1 in vitro [115] and in vivo—in cortical and hippocampal neurons by up to 44% and 24%, respectively [28,70,95]. BACE1 is the rate-limiting enzyme that catalyzes the proteolytic cleavage of APP into Aβ [27]. Thus, RJ inhibition of BACE1 indicates its ability to block the initial pathogenic steps underling the formation of Aβ plaques, the main pathological structure of AD. On the other hand, RJ significantly increased the expression of some Aβ-degrading enzymes such as insulin-degrading enzyme (IDE) [28] and neprilysin (NEP) [116]. These enzymes convert Aβ polypeptide into benign forms [28]. RJ also promoted Aβ clearance by upregulating the expression of LRP-1, which facilitates the removal of degraded Aβ out of brain cells [28,95].

RJ has been used to activate cAMP/PKA and CREB-dependent signaling linked to CRE-mediated transcription in PC12 cells via the ERK/MAPK signaling cascade [93,130]. CRE-mediated transcription is a principal signaling pathway that promotes learning and memory through long-term hippocampal potentiation. Dysregulation of this pathway influences Aβ metabolism and stimulates tau phosphorylation, which contributes to the development of cognitive impairment in neurodegenerative diseases such as AD [80,93]. Several lines of evidence note that RJ repairs dysregulated neurons encompassing Aβ pathology via activation of CREB signaling [28,116]. In one study, intragastric administration of RJ (300 mg/kg/day) to ten-month-old APP/PS1 transgenic mice for three months markedly ameliorated cognitive deficits and decreased soluble and insoluble Aβ40 and Aβ42 (25%, 40%, and 60%), as well as the size and number of SPs, both in the cortex and hippocampus, through the activation of cAMP, p-PKA, p-CREB and BDNF [28]. Likewise, another in vitro study reported that a DMSO-soluble fraction of RJ increased the clearance of soluble Aβ through the activation of the NEP-somatostatin system in hippocampal neurons. Chromatin immunoprecipitation-qPCR assays revealed that RJ enhanced the gene expression of NEP and somatostatin in hippocampal neurons. The former is a basic degrading enzyme of Aβ oligomers, while the latter facilitates NEP-mediated proteolytic degradation in the hippocampus. RJ facilitated CREB-binding to the prototypical functional CRE at the promoter region of somatostatin. The activation of the hippocampal NEP-somatostatin system plays an important role in memory formation, synaptic plasticity, neural development, and neuronal circuit homeostasis [116].

Downregulation of IIS accounts for another mechanism employed by RJ to reduce amyloidogenesis and proteotoxicity [92]. In aged *C. elegans*, RJ downregulated daf-2, a key upstream component of IIS, resulting in phosphorylation of three core downstream transcription factors of IIS: DAF-16—the *C. elegans* counterpart to the mammalian FOXO, heat shock transcription factor 1 (HSF-1), and SKN-1/NRF2 [92]. These transcription factors represent specific longevity pathways that have the potential to activate multiple cascades essential for antioxidant and anti-inflammatory activities, DNA repair, autophagy, stress resistance, and cell proliferation [63]. On the other hand, cumulating knowledge indicates that aging-related physiological alterations increase the production of highly toxic Aβ fibrils via attenuation of HSF1 signaling [194]. Knock down of DAF-16, SKN-1, and HSF-1 genes via RNA interference abolished the effect of RJ/eRJ on paralysis induced by Aβ toxicity in *C. elegans*, which confirms the main role of these genes in RJ protection against neurodegeneration [92]. Emerging knowledge highlights the role of interventions that foster the activity of FOXO and HSF-1 in hindering or possibly preventing the onset of neurodegeneration [194].

### 7.8. Royal Jelly Alleviates Hormonal and Metabolic Abnormalities Underlying Cognitive Impairment

Accumulating knowledge shows that sex steroids (estrogens, androgens, and luteinizing hormone) possess strong anti-inflammatory and neuroprotective effects. They interact with many pathways involved in AD pathology to improve insulin resistance and enhance adaptive programs that promote DNA repair capacity of the CNS. Age-related declines in these hormones contribute to the development of cognitive impairments and even AD in people with genetic and environmental vulnerabilities [24,195,196]. Estrogen supplementation has been used to improve cognitive performance and alleviate pathological damages embedded in various physical disorders (e.g., cardiovascular diseases) in reproductively-senescent women. However, estrogen as a supplement may cause devastating effects such as increased risk of cancer [63,87]. Therefore, research has been focused on the use of natural agents such as foods rich in phytoestrogens as safe substitutes for estrogen. Phytoestrogens are similar to estrogen chemically, structurally, and functionally [85,87].

RJ is reported to improve testosterone, dehydroepiandrosterone sulfate, and estradiol in healthy adults [197,198]. It is widely used to treat infertility because of its phytoestrogen content [87,199]. Thus, it is possible that RJ-related improvement of gonadal function may represent an endocrine modulation mechanism for enhancing cognitive function in old age. In fact, fatty acids and sterols in RJ, e.g., 10-HDA and 24-methylenecholesterol, exert neurogenic effects and improve brain function through estrogen signaling [95,143]. Estrogen receptor 1 represents a common single nucleotide polymorphism that contributes to the onset of AD, as indicated by genome-wide association studies [200]. The binding of RJ fatty acids to estrogen receptors β and α modulates cell proliferation, increases the production of neurotrophins such as BDNF and NGF, regulates the expression of various genes that counteract inflammation and oxidative stress in cholinergic neurons, promotes Ca^2+^ outflow and ACh release, reduces tau protein phosphorylation, decreases Aβ production, and inhibits Aβ-related destructive effects on cellular homeostasis [85,87,201]. These effects result from multiple distinct mechanisms: (1) estrogen translocation into the nucleus to bind estrogen response elements (EREs) located in the promoters of target genes, (2) facilitating protein–protein interactions with other DNA-binding transcription factors in the nucleus, and (3) nongenomic actions of estrogen that involve modifying functions of cytoplasmic proteins leading to the regulation of gene expression [143,202]. Experimental modulation of estrogen by supplementing ovariectomized animals with RJ was shown to significantly enhance cognitive behaviors, improve autonomic and cardiovascular conditions, and be capable of treating pathologies associated with AD such as Aβ deposition and hypercholesterolemia [87,95].

A strong correlation between thyroid dysfunction and AD was demonstrated by 14 out of 23 studies [203]. Research documents an association of thyroid stimulating hormone and free triiodothyronine (T3) with regional cerebral blood flow in patients with MCI and AD [204]. Deficiency of T3 and thyroxine (T4) promotes oxidative stress and alters cellular metabolism, which impair the function of all organs and contribute to metabolic and cardiovascular disorders, increasing the risk of AD. On the other hand, hypothyroidism impairs neurotransmitter synthesis, induces hyperphosphorylation of tau proteins, promotes glutamate excitotoxicity, reduces synaptic plasticity, inhibits hippocampal neurogenesis, and promotes hippocampal neuron death, which result in symptoms of poor memory and concentration [5,73]. RJ supplementation (100 mg/kg/day/20 days) to rat models of hypothyroidism (induced by a daily intraperitoneal injection of 10 mg/kg propylthiouracil) increased serum T4, reduced vascular dilation and neurodegeneration in CA3 and CA1 hippocampal regions, and increased hippocampal MAP-2 levels [73]. These findings suggest that RJ prevents neurodegeneration by maintaining microtubule and axon stability and decreasing tau phosphorylation through microtubule affinity regulating kinase signaling [73,80].

Cholesterol promotes the formation of Aβ plaque and neuronal loss by accelerating the cleavage of APP [95]. Research also indicates that high fat diets impair memory by inhibiting the signaling associated with the expression of genes involved in memory, as well as by evoking several other drastic disturbances: increased triglyceride stores, impaired metabolism, hypertension and other cardiovascular disorders, obesity, and shortened life span [205]. Two studies included in this review treated ovariectomized animals with a high cholesterol diet to induce cognitive impairment [70,95]. RJ and peptides in pRJ exhibit antihypertensive and anticholesterol effects in rabbit models of AD and in rats with high spontaneous blood pressure, such as: inducing aortic relaxation; improving heart rate variability and baroreceptor sensitivity; and reducing blood pressure, plasma lipids (e.g., TC and LDL-C), body weight, and brain levels of Aβ, AchE, and MDA [70,206,207]. Such effects were associated with increased antioxidative capacities, amelioration of Aβ pathology, and protection against neuronal damage [70]. The suggested mechanism entails agonism of muscarinic receptors by ACh in RJ, leading to a brief increase in the levels of NO and cyclic guanosine monophosphate, as well as suppression of norepinephrine-related intracellular Ca^2+^ release and K^+^-related extracellular Ca^2+^ influx [206].

## 8. Discussion

The results of this review indicate that RJ can enhance cognitive performance and boost learning and memory in old age and in AD models. Most studies promisingly signify a general trend of positive dose-dependent effects of RJ on the pathologies underlying cognitive impairments (e.g., oxidative stress, neuroinflammation, and Aβ and tau pathology) [28,70,92,95,115,116,119] in vitro and in vivo, despite the large heterogeneity in cell lines and species that were used as models of AD or cognitive aging, as well as variations in RJ preparation (e.g., enzyme treatment, lyophilization, etc.), dose, and route of administration (oral, gavage, or subcutaneous) across the available literature. Additionally, RJ as a monotherapy for AD has not been tested in a single human trial until now. Nonetheless, a few studies have investigated the effect of herbal mixtures containing RJ on cognitive function in humans. The methodological flaws in these studies give rise to serious concerns, e.g., small sample size, lack of assessment of biological markers, lack of blinding, and absence of a comparison group, to name a few [125,126].

The effects of RJ on cognition are closely associated with the modulation of various biological activities that boost the structure and operative performance of both neurons and non-neuronal brain cells (e.g., glial cell and HBMECs), and eventually enhance cell survival and prevent neurodegeneration [28,112,113,114]. Among these pharmacological activities, RJ regulated the production of neurotransmitters such as serotonin [107,119,121] and neurotrophins such as BDNF and NGF, which protect against synaptic loss in the cortex and hippocampus [28]. It was also shown to deactivate cellular-stress signaling pathways engaged in inflammation, oxidative stress, and mitochondrial-related apoptosis [28,70,119,208]. In addition, it ameliorated the detrimental effects of Aβ by decreasing its production, sequestering its insoluble form, and enhancing its removal out of the brain [28,92,95,115,116]. Additionally, RJ improved cognitive behavioral deficits in AD by correcting systemic pathologies contributing to neurodegeneration, e.g., enhancing estrogen levels, improving autonomic, metabolic, and cardiovascular conditions in ovariectomized animals [95], and increasing fT4 levels in hypothyroidism rodents model of cognitive dysfunction [73]. Therefore, future RCTs should explore if RJ can comprehensively optimize metabolic and endocrine parameters in individuals enduring age-related cognitive impairments by expanding standard laboratory evaluations.

### 8.1. Royal Jelly May Improve Health and Extend Lifespan in Cognitively Impaired Subjects

Aging is a key risk factor for AD. Despite the long lifespan of people with AD, they suffer prolonged disability out of the dramatic progress of the disease and a variety of other aging-related comorbidities [25,60,65]. As illustrated before, AD pathology may result from other existing metabolic and hormonal dysregulations such as hyperlipidemia, obesity, type 2 diabetes mellitus, hypertension and other cardiovascular disorders, as well as severely decreased levels of sex steroids [5,13,54,58]. Research shows that Aβ, tauopathy, and mutations of genes contributing to AD, e.g., Ankyrin 1 gene increase neurodegeneration, disrupt memory, decrease locomotion, and shorten lifespan in *Drosophila* [209]. We have previously shown that RJ modulates several aging pathways to prolong lifespan in various model organisms and improve health both in humans and animal models [63]. The literature examined in the current review indicates that RJ brings about other benefits along with improved cognitive function. In this regard, RJ positively affected lipid profiles in postmenopausal women (who are at high risk for AD) [126] and in AD models induced by high cholesterol diet on top of ovariectomy or copper intoxication [70,95]. It also improved estrogen levels; estrogen expresses a range of health promoting activities, e.g., promoting cardiovascular function and bone density [210,211]. More, cognitive recovery resulting from RJ treatment in hypothyroidism models was associated with improvement of plasma levels of fT4 [73]. Evidence from a current *Drosophila* model shows that RJ improved food intake, body weight, exercise capacity, and mean lifespan in flies treated with H_2_O_2_ or paraquat, as well as in naturally aged flies [96]. Taken together, these reports suggest that RJ might improve health, optimize lifespan, and promote overall wellness in cognitively impaired people by targeting age-related physiological declines. Nonetheless, studies addressing the effect of RJ on cognition in old age should be cost-benefit oriented, taking into account restoration of overall function and reduction of morbidity and disability.

AD patients experience other neuropsychiatric problems such as mood dysregulation and depressive symptoms, especially during the prodromal stage [25]. RJ may alleviate the depression that cooccurs in AD. Reports revealed that RJ at high and low doses (400 mg/kg and 10 mg/kg) reduced depression-like behavior in ovariectomized rat and rabbit models of AD to the levels of sham-operated animals [87,95]. Similarly, a combination of RJ and floral pollens resulted in significant amelioration of depressive symptoms, along with improvement of problem-solving capacity in postmenopausal women [126,127]. On top of this, the literature showed that RJ has the potential to improve general mental health of healthy volunteers [197]. It also documents antidepressant effects of RJ in animal models of depression and anxiety [201,212]. The cellular and molecular mechanisms underlying the action of RJ in animal models of AD and cognitive aging is not completely understood. However, it seems that RJ affects common pathologies in AD and depression. For instance, RJ improved levels of serotonin (a key neurotransmitter involved in the pathogenesis of depression), both in naturally aged rats [107,110] and AD models induced by cortical neuron death following intoxication with cadmium and tartrazine [119,120]. It also alleviated microglial inflammation in LPS-stimulated BV-2 cells as an in vitro model of AD [113,117]. Microglial activation is evident in the pathology of depression as well [213]. Therefore, RJ treatment of cognitive impairment may yield multipartite benefits by affecting existing mood pathologies. Nonetheless, soundly designed investigations are needed to examine the effect of RJ on cognitive dysfunction and related depressive states in AD.

### 8.2. Tips for Identifying Treatment Targets

At present, universally accepted biomarkers and preventive strategies of AD are quite limited [214]. Imaging studies state that an extremely long preclinical phase paves the way for the active symptomatic phase of AD. The microbiome alterations, oxidative stress, chronic inflammation, and metabolic dysregulations that occur during the preclinical phase promote gradual buildup of Aβ pathology and tauopathy associated with sporadic AD. These pathologies may start to develop in the lower brainstem early, i.e., at the age of 40 [20,33]. MCI progresses to AD dementia at an annual rate of 7%, which is quite alarming. Therefore, detecting and treating cognitive decline at an early stage is necessary to prevent the development of dementia [204]. Attention has been focused on the use of simple, low-cost interventions to manage modifiable risk factors for AD, such as metabolic dysfunctions (e.g., diabetes mellitus, hypertension, hyperlipidemia, and obesity), sedentary life style, and unhealthy diet [214]. Depressive disorders, especially treatment resistant ones, may represent another AD prevention target. A long-term, large-scale cohort study reporting data on psychiatric symptoms in 1998 participants who progressed to MCI or dementia indicated that depression and irritability are the most common psychiatric symptoms occurring before cognitive diagnoses (24% and 21%, respectively) [25]. Depression cooccurring with AD is not considered clinical depression, but rather, an indicator of AD progression, i.e., it is an outcome of severe neurodegeneration, which makes it highly resistant to the current best recommended treatment for depression [215].

Because AD is multifactorial, studies should focus on understanding the effect of RJ on various risk factors and the mechanisms underlying cognitive protection under such circumstances. Research documents the formation of Aβ plaques in other disorders with PS1 mutations such as epilepsy. Furthermore, people with such disorders are likely to develop AD later in life [36]. In the same way, the potential effects of endocrine disturbances (e.g., thyroid dysfunction and sex steroid deficiency in senescent individuals) on the molecular mechanics of AD pathology are becoming increasingly acknowledged [195,196]. Indeed, natural estrogen agonists are now being used to inhibit the destructive effects of Aβ in AD [85]. Thus, proper screening and management of cognitive alterations in menopausal women and other endocrine disorders may be an important prevention strategy of AD. From another perspective, the GI leak that occurs in GI disorders such as irritable bowel syndrome represents a gate that continually influx bacterial toxins into the blood stream to produce a persistent inflammatory response in the CNS. Therefore, efforts directed toward eliminating or neutralizing GI bacterial toxins may abort the inflammatory reactions contributing to AD [41].

Due to their antimicrobial properties, RJ, 10-HDA, MRJPs, and other proteins (e.g., royalisin, jelleines, and aspimin) may alleviate GI dysfunction and prevent dysbiosis by inhibiting enterotoxic pathogens [63]. RJ and other bee products contain high levels of various strains of lactic acid bacteria, which can inhibit numerous antibiotic-resistant human pathogens [216]. Several lines of evidence show that RJ increases lactic acid production and fosters the growth and proliferation of probiotic bacteria such as *Bifidobacterium animalis* spp. *Lactis* and *Lactobacillus acidophilus*, as well as other gut bacteria such as *Bacteroides thetaiotaomicron*, which take part in the activation of the regulatory T cells [217,218]. In this respect, research denotes that rats with GI injury and imbalanced gut microbiota exhibit memory impairment. Meanwhile, supplementing *Lactobacillus johnsonii* to rats suffering from colitis (induced by ingestion of toxic chemicals or fecal pellets rich in *E. coli* from rats with colitis) corrected the composition of gut microbiota, decreased gut and blood levels of LPS, and counteracted memory impairment induced by colitis [40]. In addition, experimental studies both in vitro and in vivo show that RJ contributes to the integrity of the intestinal wall and protects against GI injury resulting from the ingestion of harmful substances or allergenic foods by ameliorating oxidative stress, inhibiting the secretory response, and increasing villus length [219,220,221]. Therefore, the effect of RJ in AD may involve a direct effect on the structure of microbiota population of the gut, which needs to be explored in future investigations. On the other hand, the activity of GI microbiota may be necessary for RJ to demonstrate its therapeutic activities. For instance, certain species of gut microbiota can actively convert dietary polyphenols into phenolic acids, which can then be passed to the blood stream to subsequently cross the BBB and interfere with the production and accumulation of Aβ in the brain [222].

### 8.3. Issues of Concern Regarding the Use of Royal Jelly in Research Related to Cognitive Aging

The available experimental and clinical trials employing RJ for the management of age-related cognitive decline provide little information on RJ elements that demonstrate the most potent anti-AD effect. The majority of the studies treated animals/cell lines with whole crude RJ, lyophilized powder, enzyme-treated, or DMSO extract [107,110,116,123]. Although positive effects of crude RJ were reported in several studies, investigations involving both crude and eRJ reported stronger effects of the latter [92,123]. This is because the treatment of RJ with bacterial enzymes increases the amount of amino acids and short peptides without altering other bioactive contents such as 10-HDA. The phenolic group in amino acids and short peptides of RJ contribute to their high antioxidant properties through electron donation [63]. A few studies have suggested that some fractions of RJ are potentially therapeutic agents for pathologies underlying memory impairment in advanced age. Evidence shows that digested peptides of RJ interfere with Aβ formation and accumulation in N2a/APP695swe microglia via downregulation of BACE1 [115]. Additionally, MRJPs are reported to positively affect brains of aged rats by regulating pathways involved in energy metabolism and oxidative stress. Whether these effects stem from a single active MRJP/its metabolites or from interactions of MRJP family proteins is not clear yet [109]. In addition, 10-HDA has been reported to protect microglia and the BBB against LPS-induced toxicity both in vivo and in vitro by inhibiting the expression of ROS, cytokines, chemokines, adhesion factors, and matrix metalloproteinases, as well as by enhancing autophagy [114,117]. It is suggested that the RJ contents of ACh and enzymes (e.g., lipase and SOD) are responsible for memory improvement [106]. The identification of the most effective anti-AD fractions of RJ is critical, should it be used as a cost-effective treatment of cognitive aging. In this respect, valuable information may stem from future research that examines the anti-AD effect of other elements derived from RJ. For instance, HPO-DAEE [144] and AMP N1-oxide [131] act as potent regulators of neurotrophin production. The latter is reported to stimulate neurite growth, neurogenesis, and astrogenesis. It also potently suppresses the inflammatory effects of LPS in RAW264.7 macrophages through the activation of PI3K/Akt/GSK-3β signaling [131,139,143].

Dosage, route of administration, and length of treatment represent areas of concern. Different doses of RJ have been used (50 to 500 mg/kg/day). However, models used to demonstrate cognitive impairments varied ranging from natural aging [107,108,110,121] to copper intoxication plus high cholesterol diet [70]. It has been stated that the effects of RJ on cholesterol occur at a dose of 400 mg/kg body weight/day, and that this dose can positively affect the brain structure in ovariectomized cholesterol-fed rabbits [95]. Yet, disrupted fat metabolism does not occur in all cases of AD, and other underlying pathologies should be taken into consideration when choosing an appropriate dose. Animals in the majority of in vivo studies were treated with RJ for a relatively long time (10 days or more) either orally or through gavage. RJ enhancement of memory was reported in all in vivo studies except for one short-term study [121], which involved subcutaneous administration of RJ (100 and 500mg/kg body weight/day) to naturally aged mice for 6 days. In spite of this, short-term RJ treatment significantly affected cortical serotonin metabolism [121]. On the other hand, 5% dietary RJ supplementation to mice with cognitive impairment associated with trimethyltin induced-acute neuronal death of hippocampal DG alleviated cognitive impairment, decreased neuronal loss, and stimulated the generation of functional hippocampal neurons [112]. The noted variations between these two studies implies a possible role of the route of administration in the effect of RJ.

Combinations of RJ with collagen peptides from fish skin [96] and various herbal plants [125,126,127] produced positive behavioral and biological effects. Therefore, comparing outcomes of RJ monotherapy and combinations of RJ with herbal extracts and other natural products may be necessary for maximizing its therapeutic effectiveness. A major limitation encountered by studies using RJ and other bee products to target cognition in old age is the clear variation in the ingredients of a single bee product according to bee species, botanical origin, season, geographical location, bee supplementation with proteins or sugars, methods of processing, etc. [63]. In the meantime, universally accepted quality standards for the production of RJ and other bee products are lacking. For instance, a comparison of 10-HDA content in RJ from Greece with samples from other countries revealed that Brazilian RJ contains higher levels of 10-HDA than that from other countries, such as Switzerland, Japan, India, and Turkey [223].

### 8.4. Safety of Prolonged Consumption of Royal Jelly in Old Age

RJ is abundant in biologically active compounds, which contribute to its diverse range of therapeutic properties and make it a leading substance for drug development. However, designing AD treatment clinically requires careful empirical testing that weighs benefits versus risks. The findings investigated in this review indicate multiple positive effects of RJ with minimal or no side effects. In this respect, the long-term use of RJ in aged rats was shown to have no negative effects on the functions of the liver and kidney, which emphasizes the safety of its long-term use in old age [122]. Notably, few side effects have been reported in studies recruiting human subjects [125,126]. However, these studies used a combination of RJ with herbal plants, and therefore, the reported side effects may not be attributable to RJ alone. Interestingly, high oral doses of RJ (20 g) significantly reduced glucose levels in healthy human volunteers without causing side effects [224]. Furthermore, the toxicity effect of RJ has been examined in BV-2 microglial cells: RJ up to a concentration of 3 mg/mL for 24 h had no cytotoxic effects [113]. Given that some ingredients of RJ such as proteins can be degraded by gastric enzymes before they reach the intestine [63], achieving a similar cellular level following oral consumption may require exceptionally high doses. From another perspective, concerns are rising regarding the therapeutic use of bee products and herbal plants due to contamination with pesticides such as neonicotinoids, which are used on a large scale worldwide. Nonetheless, research has documented trivial concentrations of neonicotinoids in RJ [63,225], i.e., equal to 0.016% of the original concentrations of pesticide fed to bee workers during experimental breeding [225].

Allergic reactions to bee products are documented in the literature. Despite the reports indicating that RJ can reduce serum histamine, IgG, and IgE levels in various allergic conditions by suppressing histamine H1 receptor (H1R) [220,224], in rare cases, some proteins in RJ may trigger an IgE anaphylactic reaction [96,100]. Nonetheless, enzyme treatment of RJ involves the removal of allergen proteins. Exceptionally, eRJ contains higher levels of bioactive ingredients such as dipeptides, tripeptides, and amino acids in larger amounts than in crude RJ, because RJ hydrolysates are converted into shorter chain monomers that are easy to absorb. Meanwhile, levels of other active ingredients such as 10-HDA are not affected [63,96,224,225].

## 9. Conclusions

The studies scrutinized in this review show that RJ has a potential to recover cognitive deficits and increase glial and neuronal cell survival in experimental models of advanced aging and AD. RJ demonstrates neuroprotection against Aβ and tau pathologies via enhancement of the antioxidant system, suppression of inflammation, augmentation of neurotrophin production and synaptic signal transduction, all of which promote healthy neuronal structure and function. Additionally, animal studies have shown that RJ can improve health and extend lifespan via enhancement of various general metabolic and endocrine parameters. Enzyme-treated RJ seems to be more effective than crude RJ, while MRJPs, peptides, and 10-HDA were the only RJ compounds examined in experiments on cognitive aging. Preclinical and clinical studies confirm the safety and efficacy of RJ treatment both in vivo and in vitro. Clinical trials evaluating the effect of RJ on cognitive performance in humans are limited both in quantity and quality. Many questions remain unanswered, and multiple issues of concern exist: proper dosage, use of RJ alone or in combination with other treatments, and confounding effects of other, related factors such as diet, activity, and race. Further rigorously designed RCTs that employ biological markers are needed to confirm the reproducibility of findings obtained from preclinical trials in human subjects.

## Figures and Tables

**Figure 1 antioxidants-09-00937-f001:**
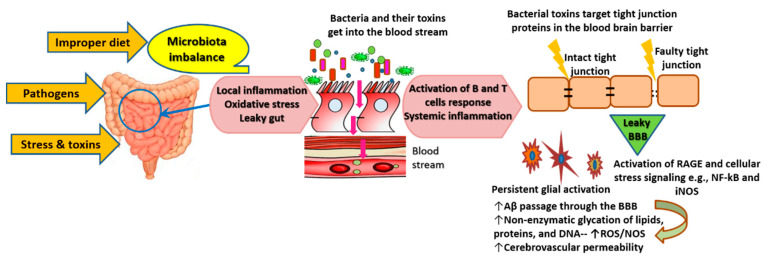
Contribution of disrupted microbiota to the permeability of the blood brain barrier and neuroinflammation in Alzheimer’s disease. Abbreviations: ↑ denotes increase, BBB: blood brain barrier, RAGE: receptor for advanced glycation end products, NF-κB: nuclear factor-kappa B, iNOS: inducible nitric oxide synthase, Aβ: beta-amyloid protein fragments, ROS: reactive oxygen species, NOS: nitric oxide species.

**Figure 2 antioxidants-09-00937-f002:**
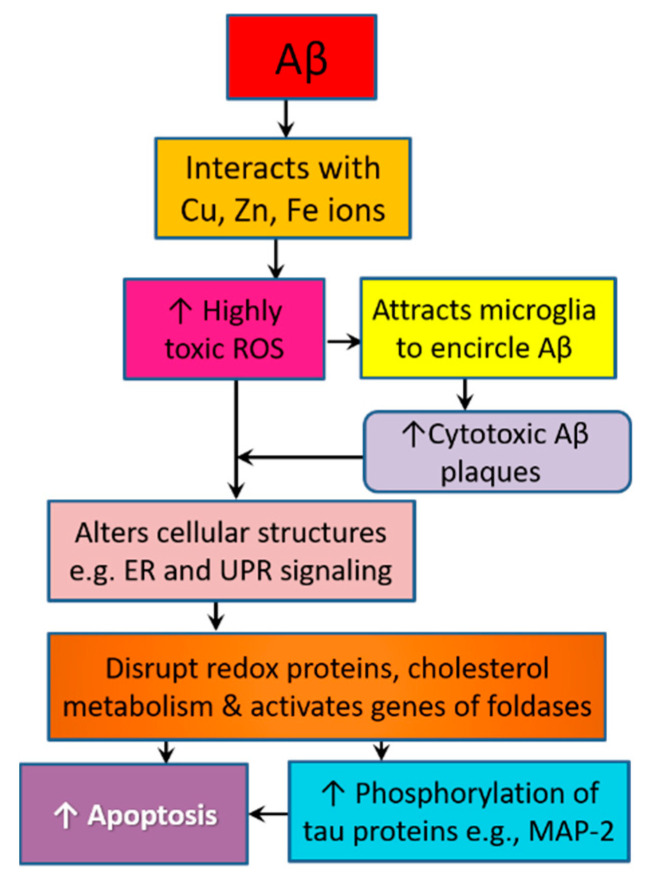
Beta-amyloid protein fragments induce multiple molecular and cellular changes that promote neurodegeneration. Abbreviations: ↑ denotes increase, Aβ: beta-amyloid protein fragments, ROS: reactive oxygen species, ER: endoplasmic reticulum, UPR: unfolded protein response, MAP-2: microtubule-associated protein 2. Aβ pathology contributes to various molecular and cellular changes that induce neuronal death.

**Figure 3 antioxidants-09-00937-f003:**
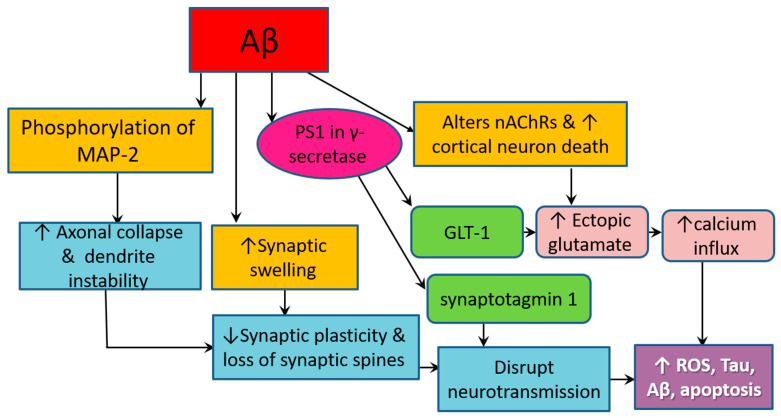
Schematic illustration of the mechanism underlying synaptic destruction by beta-amyloid peptide (Aβ). **Abbreviations**: ↑ denotes increase, ↓ denotes decrease, Aβ: beta-amyloid protein fragments, MAP-2: microtubule-associated protein 2, PS1: presenilin 1, nAChRs: nicotinic acetylcholine receptors, GLT-1: glutamate transporter 1, ROS: reactive oxygen species. Aβ pathology contributes to synaptic destruction, which promotes neuronal apoptosis through various mechanisms.

**Figure 4 antioxidants-09-00937-f004:**
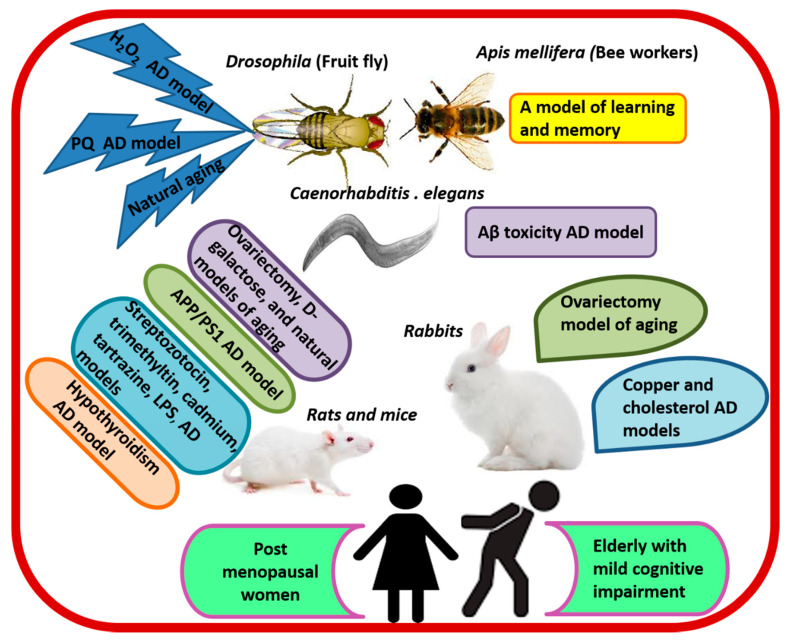
Royal jelly improves cognitive function-related parameters in various animal models and in humans. Abbreviations: AD: Alzheimer’s disease, PQ: paraquat, H_2_O_2_: hydrogen peroxide, Aβ: beta-amyloid peptide, APP/PS1 mice express 2 mutations associated with early-onset AD—chimeric mouse/human APP (Mo/HuAPP695swe) and human PS1 (PS1-dE9), LPS: lipopolysaccharide.

**Figure 5 antioxidants-09-00937-f005:**
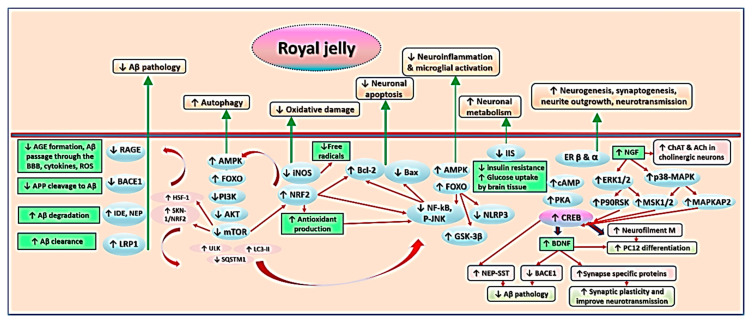
Possible mechanisms of action behind royal jelly-related effects in cognitive aging and Alzheimer’s disease. Abbreviations: ↑ denotes increase, ↓ denotes decrease, Aβ: beta-amyloid protein fragments, RAGE: receptor for advanced glycation end products, BBB: blood brain barrier, ROS: reactive oxygen species, APP: amyloid precursor protein, BACE1: beta-site APP cleaving enzyme 1, IDE: insulin-degrading enzyme, NEP: neprilysin, SST: somatostatin, LRP-1: low density lipoprotein) receptor-related protein 1, iNOS: inducible nitric oxide synthase, NRF2: nuclear factor-erythroid 2-related factor 2, AMPK: AMP-activated protein kinase, FOXO: Forkhead Box O transcription factor, HSF-1: heat shock transcription factor 1, PI3K: phosphatidylinositol 3-kinase, PKA: cAMP-dependent protein kinase, cAMP: cyclic adenosine mono phosphate, CREB: cAMP-response element (CRE)-binding protein, mTOR: mammalian target of rapamycin, ULK: Unc-51-like kinase, LC3-II: microtubule-associated protein 1 light chain 3-II, SQSTM1: sequestosome 1, Bcl-2: B-cell lymphoma-2, Bax: Bcl-2-associated X protein, NF-κB: nuclear factor-kappa B, JNK: c-Jun NH2-terminal kinases, NLRP3: nucleotide-binding domain and leucine-rich repeat containing protein 3, GSK-3β: glycogen synthase kinase-3β, IIS: insulin/insulin-like growth factor, ER β and α: estrogen receptors β and α, MAPK: mitogen-activated protein kinase, ERK1/2: extracellular signal-regulated kinase 1 or 2, NGF: nerve growth factor, ACh: acetylcholine, ChAT: choline acetyltransferase, BDNF: brain derived neurotrophic factor, p90RSK: pp90 ribosomal S6 kinase, MSK1/2 and MAPKAP: mitogen- and stress-activated protein kinase and kinase 2, PC12: progenitor stem cells.

**Table 1 antioxidants-09-00937-t001:** Composition of royal jelly.

Compounds	Percentage	Examples	References
Moisture	50–70%	Mainly water.	[96,98,99]
Carbohydrates	7.5–16%	Sugars such as fructose, glucose, maltose, melibiose, and ribose.	[98,99]
Proteins	9–18%	MRJPs (80% of protein content), minor proteins (e.g., aspimin, royalisin and jelleines), peptides (in the form of dipeptides or tripeptides, e.g., alanine-leucine, leucine- aspartic acid-arginine), and free amino acids (e.g., threonine, valine, glycine, isoleucine, leucine, proline, serine, methionine, and tryptophan).	[63,99,102]
Lipidsa	3–6%	10-HDA, sebacic acid, phenols (4–10%), waxes (5–6%), steroids (3–4%), and phospholipids (0.4–0.8%).	[63,98,99,101,102]
Vitamins	?	B5 (52.8 mg/100 g), B6 (42.42 mg/100 g), niacin (42.42 mg/100 g), and traces of B1, B2, B8, B9, B12, ascorbic acid (vitamin C), vitamin E and A.	[63,102]
Minerals	?	Potassium, sodium, magnesium, calcium, phosphor, sulfur, cupper, iron, zinc, selenium, barium, cobalt, manganese, etc.	[42,63,98]
Bioactive compounds	?	ACh and nucleotides both as free bases (e.g., adenosine, guanosine, iridin, and cytidine) and as phosphates (e.g., adenosine 5′-monophosphate, adenosine 5′-diphosphate, and adenosine 5′-triphosphate).	[63,102]
Others	?	Volatile organic compounds (e.g., esters, aldehydes, ketones, and alcohols), and minor heterocyclic compounds.	[63,98,102]

**Abbreviations**: MRJPs: major royal jelly proteins, 10-HDA: Trans-10-hydroxy-2-decenoic acid, ACh: acetylcholine. N.B. percentages of constituents are reported in fresh RJ, and data on the percentages of RJ constituents denoted by “?” are not clearly available.

**Table 2 antioxidants-09-00937-t002:** Effects of royal jelly on cognition and related molecular-level changes in experimental models and humans (N of reviewed studies = 30).

Animal/Cell Line Model	RJ Treatment	Summary of Effects and Mechanism	Reference
Hippocampal SST and NEP positive neurons	DRJ (100 mg/mL)	↑SST and NEP gene expression and CREB-binding to CRE at the promoter region of SST.	[116]
N2a/APP695 cells	RJPs (1–9 μg/mL)	↓Aβ1-40, Aβ1-42, and BACE1.	[115]
LPS-stimulated BV-2 microglia	RJ (0.3–3 mg/mL)	↓IL-6, IL-1β, TNF-α, iNOS, and COX-2.	[113]
*Apis mellifera* workers as a model of learning	RJ (10% and 20%) in 50% sucrose solution	↑Olfactory learning, memory, and expression of memory-related genes (GluRA and Nmdar1).	[105,106]
Naturally aged *Drosophila* and *Drosophila* treated with H_2_O_2_ and paraquat	eRJ (1–5 mg/mL) plus CP at a ratio of 2:3	↑T-SOD, GSH-Px, CAT, average life span, food consumption, weight gain, and exercise capacity. ↓MDA and protein carbonyl.	[96]
Aβ toxicity in CL2006 worm model of AD	RJ (2 mg/mL) and eRJ 1 mg/mL)/day/10 days at 20 °C	↓Aβ species, Aβ-induced body paralysis, and IIS signaling. ↑Soluble proteins.	[92]
LPS-stimulated C57BL/6J mice and microglial BV-2 cells	Oral 10-HAD (100 mg/kg/day for 1 month)	↓TNF-α, Tnfrsf8, Traf1, IL-1β, NF-κB and NLRP3 inflammasome-IL-1β signaling, and SQSTM1. ↑FOXO1-mediated autophagy, ULK, and LC3-II.	[117]
LPS-stimulated C57BL/6 mice and HBMECs	Oral 10-HAD (100 mg/kg/day for 1 month)	↓ROS, CCL-2, CCL-3, ICAM-1, VCAM-1, MMP-2, and MMP-9, BBB permeability, and tight junction proteins degradation. ↑Expression of tight junction proteins, and AMPK/PI3K/AKT signaling.	[114]
OVX cholesterol-fed rabbit model of AD	Oral RJ (400 mg/kg/day/12 weeks)	↓Behavioral cognitive deficits, body weight, blood lipid, BBB permeability, brain levels of MDA, Aβ, AchE, BACE1, and RAGE. ↑ChAT, SOD, LRP-1, heart rate variability, and Baroreflex sensitivity.	[95]
OVX rat model of aging	Oral eRJ (250 mg/mL tap water: 10 mL/kg/day/82 days)	↓Cognitive and depressive-like behavioral deficits. ↑Brain weight and myelin galactolipids.	[87]
A rat model of AD induced by streptozotocin (icv)	Oral RJ (200 mg/kg/day/14 days)	↓O_2-_ (in the DG and hilus regions) and neurodegeneration (in the DG). ↑Working memory and neurogenesis in the DG.	[111]
Hypothyroidism rat model of cognitive impairment	Intragastric RJ (100 mg/kg/day/20 days)	↓Brain vascular dilation, edema, and degeneration. ↑MAP-2 and fT4.	[73]
APP/PS1 transgenic mice model of AD	Intragastric RJ (300 mg/kg/day/3 months)	↑Spatial learning and memory. ↓MDA, p-JNK and bax/bcl-2 ratio, caspase-3, BACE1, Aβ40 and Aβ42, and the total area and number of senile plaques in the cortex and hippocampus. ↑cAMP, p-PKA, p-CREB, BDNF, IDE, and LRP-1.	[28]
A rabbit model of AD induced by cholesterol diet and copper sulfate	Oral RJ (400 mg/kg/day/12 weeks)	↓TC, LDL-C, MDA, ROS, RNS, Cho/Cr, mI/Cr, caspase-3, BACE1, Aβ1-40, Aβ1-42, Aβ plaque, and neuronal loss. ↑SOD, LRP-1, IDE, NAA/Cr, and glutamate/Cr.	[70]
A mouse model of streptozotocin-induced cognitive impairment	Dietary RJ (3% w/w/day/10 days)	↓Streptozotocin-induced defects in learning and memory.	[16]
A mouse model of trimethyltin-induced hippocampal DG damage	Dietary RJ (1% or 5% w/w/day/6 days)	↓Cognitive impairment and neuronal cell loss. ↑Number of hippocampal DG granule cells.	[112]
A mouse model of cadmium-induced cortical damage	Intragastric RJ (85 mg/kg/day/7 days)	↑NRF2, GSH-Px, GSH-R, SOD, CAT, Bcl-2, norepinephrine, dopamine, and serotonin. ↓iNOS, ROS, NOS, TNF-α, IL-1β, Bax, caspase-3, and cadmium level in cortical neurons.	[119]
A mouse model of tartrazine-induced cortical damage	Ora RJ (300 mg/kg/day/30 days)	↑ CAT, SOD, GSH, and brain levels of GABA, dopamine, and 5HT. ↓MDA, cortical pyknotic nuclei, and ssDNA positive apoptotic cells.	[120]
Naturally aged rats	Oral RJ (50 and 100 mg/kg/day/8 weeks)	↑Memory and learning.	[107]
Naturally aged rats	Dietary RJ (3% w/w/day/10 days)	↑Memory and learning.	[108]
Naturally aged rats	Intragastric MRJPs (125 mg/kg/day/14 weeks)	↑Learning, memory, gluconeogenesis, brain glucose supply and ATP level, nicotinate and nicotinamide metabolism—NaMN, and cysteine-taurine metabolism. ↓ROS, AKT, and GABA.	[109]
Naturally aged rats	Oral/intragastric RJ (50 and 100 mg/kg/day/8 weeks)	↑Learning, spatial memory, and motor performance. ↓5-HT, dopamine, MHPG and its turnover. ↑5HIAA, DOPAC and their turnover in the prefrontal cortex. ↑DOPAC and ↓5HIAA in the striatum.	[107,110]
Naturally aged rats	Subcutaneous RJ (100 and 500 mg/kg/day/6 days)	↑ Serotonin activity in the hippocampus and prefrontal cortex.	[121]
Naturally aged rats	Intragastric RJ (50 and 100 mg/kg/day/8 weeks)	↓GABA in the striatum and hypothalamus.	[122]
d-galactose induced mouse model of aging	Intragastric RJ and eRJ (0.7 and 1.4 mg/kg/day/90 days)	↓ROS and body weight loss. ↑Memory, learning, muscular performance, and levels of internal antioxidant enzymes.	[123]
d-galactose induced mouse model of aging	Intragastric RJ (0.7 and 1.4 mg/kg/day/90 days)	↑Spatial learning, memory, brain levels of norepinephrine, dopamine, and SOD. ↓MDA.	[124]
Elderly with MCI	RJ plus herbal extracts	↑Scores of the Mini-Mental State Scale.	[125]
Postmenopausal women with menopausal complaints	RJ plus flower pollen	↑Problem-solving ability, HDL, and TG. ↓Depression, menopausal symptoms, TC, and LDL.	[126]

Abbreviations:↑ denotes increase, ↓ denotes decrease, RJ: Royal jelly, eRJ: enzyme-treated RJ, MRJPs: major RJ proteins, RJPs: purified RJ peptides, SST: somatostatin, NEP: neprilysin, cAMP: cyclic adenosine monophosphate, CREB: cAMP-response element (CRE)-binding protein, DRJ: DMSO-soluble fraction of RJ, Aβ: amyloid-β peptide, APP: amyloid precursor protein, N2a/APP695 cells: an in vitro model of AD pathology being stably transfected with the human APP gene to produce high levels of Aβ, DG: dentate gyrus granule, BACE1/β-secretase: beta-site APP cleaving enzyme 1, IL-6: interleukin-6, IL-1β: interleukin-1β, TNF-α: tumor necrosis factor alpha, icv: intracerebroventricular injection, iNOS: inducible nitric oxide synthase, COX-2: cyclooxygenase-2, NF-κB: nuclear factor-kB, CP: collagen peptide, T-SOD: total superoxide dismutase, GSH-Px: glutathione peroxidase, CAT: catalase, IIS: insulin/insulin-like growth factor, NLRP3: nucleotide-binding domain and leucine-rich repeat containing protein 3, SQSTM1: Sequestosome 1, ULK: Unc-51-like autophagy activating kinase, LC3-II: Microtubule-associated protein 1 light chain 3-II, TNFRSF8: tumor necrosis factor receptor superfamily, member 8, TRAF1: TNF receptor-associated factor 1, 10-HDA: 10-hydroxy-decanoic acid, HBMECs: human brain microvascular endothelial cells, ROS: reactive oxygen species, RNS: reactive nitrogen species, CCL-2: C-C motif ligand 2, CCL-3: C-C motif ligand 3, ICAM-1: intercellular adhesion molecule 1, VCAM-1: vascular cell adhesion molecule-1, MMP: matrix metalloproteinase, BBB: blood brain barrier, AMPK: 5′-AMP-activated protein kinase, PI3k: phosphoinositide-3 kinase, ATP: Adenosine triphosphate, AKT: a serine/threonine nutrient sensing protein kinase of the PI3k family, OVX: ovariectomized, AchE: acetylcholinesterase, RAGE: receptor for advanced glycation end products, LRP-1: low density lipoprotein receptor-related protein 1, MDA: Malonaldehyde, MAP-2: microtubule-associated protein 2, fT4: free thyroxine, p-JNK: Phosphorylated p-Jun *N*-terminal kinase, NRF2: nuclear factor erythroid 2, Bcl-2: B-cell lymphoma 2, Bax: Bcl-2-associated X protein, PKA: adenosine A2A receptor-mediated protein kinase A, BDNF: brain-derived nerve factor, IDE: insulin-degrading enzyme, TC: total cholesterol, LDL-C: low density lipoprotein C, HDL: high-density lipoproteins, TG: triglycerides, NAA: *N*-acetyl aspartate, Cho: choline, mI: myo-inositol, Cr: creatine, GABA: gamma-aminobutyric acid, 5HT: serotonin transporter, 5HIAA: 5-hydroxyindoleacetic acid, DOPAC: 3,4-dihydroxyphenylacetic acid, MHPG: 3-methoxy-4-hydroxyphenylglycol, NaMN: nicotinic acid mononucleotide.

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
