# Peer review of "Royal Jelly as an Intelligent Anti-Aging Agent—A Focus on Cognitive Aging and Alzheimer’s Disease: A Review"

_antioxidants, 2020, doi:10.3390/antiox9100937_

Round 1

Reviewer 1 Report

General comments:

This manuscript consists of 49 numbered pages cited 225 references. The references are properly collected and arranged. In formal aspect it is an appropriate work, and meets the requirements determined by the journal.

Specific comments:

The objective of this manuscript is to review the Alzheimer’s disease and the process of cognitive aging then explain the function and the effect of Royal jelly in the treatment of Alzheimer’s disease.

Prepare a review about cognitive disorders focusing on Alzheimer’s disease is a very difficult objective. The authors tried to solve this hard work efficiently. In the first part, the cognitive disorders was summerized, then an overview about Alzheimer’s disease was made.

The review is very precise work, the authors cited many references and discussed them logically. They presented the Royal jelly which is a natural product in the treatment of cognitive aging. The figures and tables are informative and understable. The structure of the manuscript is logical. It is written very clearly with an organizational structure that follows the logic of subject material discussed.

I found some spelling mistakes i.e. 107 line: Alzhimer instead of Alzheimer.

Royal jelly is a natural product containing many active components. The authors described the component, however it is a bit confused. I suggest that the authors have to prepare a table summerizing the active components of Royal jelly in order to characterize the active components of RJ.

Author Response

Manuscript ID: antioxidants-932302

Royal jelly as an intelligent anti-aging—a focus on cognitive aging and Alzheimer's disease: a review

Response to Comments of Reviewer 1

We thank Reviewer 1 for his/her productive and insightful comments. The comments are addressed line-by-line as shown below. Replies come underneath in red.

I found some spelling mistakes i.e. 107 line: Alzhimer instead of Alzheimer.

We apologize for that. The word “Alzhimer’s” has been replaced by the correct word “Alzheimer’s” (line 107).

Royal jelly is a natural product containing many active components. The authors described the component, however it is a bit confused. I suggest that the authors have to prepare a table summerizing the active components of Royal jelly in order to characterize the active components of RJ.

Based on this comment, Table 1 has been added to give an in-depth illustration of RJ structure (line 482).

In final, we thank Reviewer 1 for the time, effort, and help provided. We hope that the comments were properly handled and that the revised version will be suitable for publication.

 Best regards,

Reviewer 2 Report

Review of a manuscript “Royal jelly as an intelligent anti-aging—a focus on 3 cognitive aging and Alzheimer's disease: a review” by AMIRA MOHAMMED ALI and HIROSHI KUNUGI submitted to “Antioxidant”, MDPI.

Aging is associated with advanced decline of overall homeostasis and deterioration of cognitive, muscular and sensory functions. The absence of efficient treatment of these alterations requests the urgent search for alternative natural remedies that can target various underlying mechanisms associated with aging. The authors summarize the data about the effect of Royal jelly - gelatinous substance secreted by young nurse worker bees as an antiaging agent and consider its influence on various processes associated with aging. This field is very important and the analysis presented in the manuscript will be in interesting for the readers of “Antioxidants”.

The following corrections should be made:

Abstract:

Lines 17-18: ”the search for other alternative natural resources that can target different underlying pathologies of AD and prevent disease occurrence…”. This is an awkward sentence, it should be rewritten as follows: ”the search for alternative natural resources that can target various underlying mechanisms of AD pathology and reduce disease occurrence”.

Line 107: 3.1. “Role of the immune system in Alzhimer’s disease”: should be corrected as “107 3.1. Role of the immune system in Alzheimer’s disease”

Lines 164-166: Figure 1. In figure 1 legend the authors should explain the meaning of all symbols, including different types of arrows.

Lines 310-312: It is not clear why the authors use sometimes Aβ and sometimes beta-amyloid protein fragments

Line 462: “which are known as major royal jelly (MRJPs)” should be written as “which are known as major royal jelly proteins (MRJPs)”

Line 464-465 ”A recent in silico study emphasizes that all proteins in RJ demonstrate the main properties of an ideal protein: aromaticity, hydrophobicity, ionizability, and hydrogen (H)-bond [97].”

This sentence sounds weird. Why the authors believe that an ideal protein should possess these properties? Should be modified or deleted.

Lines 480-481: ”For instance, feeding bees with sugars causes significant alterations in RJ content of amino acids (e.g., tryptophan and lysine)… should be rewritten as follows :”For instance, feeding bees with sugars causes significant alterations in amino acid content of certain amino acids in RJ for example, tryptophan and lysine.

Line 499: “The exceptional cognitive abilities of queen bees are associated with persistent expression of Dnmt3gene and DNA methylation…”.

The sense of this sentence is not completely clear. Should be corrected as follows: ”The exceptional cognitive abilities of queen bees are associated with persistent expression of Dnmt3 gene encoding DNA methyltransferase – enzyme catalyzing DNA methylation”.

Line 544-545 “mitochondrial-related apoptosis [24, 67, 116]” The author need to add the following reference here: ”Effect of g-synuclein silencing on apoptotic pathways in retinal ganglion cells. J Biol Chem, 2008, 284, (52): 36377-85.

Overall this is an interesting, stimulating and thought-provoking review

Author Response

Manuscript ID: antioxidants-932302

Royal jelly as an intelligent anti-aging—a focus on cognitive aging and Alzheimer's disease: a review

Response to Comments of Reviewer 2

We appreciate Reviewer 2’s thoughtful reading and concerns for clarity as indicated by the provided comments. The comments are addressed line-by-line as shown below. Replies come underneath in red.

 Abstract:

Lines 17-18: ”the search for other alternative natural resources that can target different underlying pathologies of AD and prevent disease occurrence…”. This is an awkward sentence, it should be rewritten as follows: ”the search for alternative natural resources that can target various underlying mechanisms of AD pathology and reduce disease occurrence”.

The sentence has been modified as the reviewer requested (line 17-18).

Line 107: 3.1. “Role of the immune system in Alzhimer’s disease”: should be corrected as “107 3.1. Role of the immune system in Alzheimer’s disease”

The word “Alzhimer’s” has been replaced by the correct word “Alzheimer’s” (line 107).

Lines 164-166: Figure 1. In figure 1 legend the authors should explain the meaning of all symbols, including different types of arrows.

Yes, we have explained the meaning of symbols in Figure 1, including different types of arrows (line 164-187).

Lines 310-312: It is not clear why the authors use sometimes Aβ and sometimes beta-amyloid protein fragments

According to our knowledge, tables and figures are exclusively independent from the manuscript; they should be self-explanatory. For this reason, titles of tables and figures do not include abbreviations. All over the manuscript, Aβ has been consistently used. Instances in which “beta-amyloid protein fragments” is used are limited to figures and tables, and they are mentioned as definitions of Aβ (e.g., Aβ: beta-amyloid protein fragments) on the list of abbreviations that follows the table or the figure.

Line 462: “which are known as major royal jelly (MRJPs)” should be written as “which are known as major royal jelly proteins (MRJPs)”

The missing word “proteins” has been added to the indicated location (line 453).

Line 464-465 ”A recent in silico study emphasizes that all proteins in RJ demonstrate the main properties of an ideal protein: aromaticity, hydrophobicity, ionizability, and hydrogen (H)-bond [97].” This sentence sounds weird. Why the authors believe that an ideal protein should possess these properties? Should be modified or deleted.

Yes, the sentence was deleted as the reviewer required (line 465).

Lines 480-481: ”For instance, feeding bees with sugars causes significant alterations in RJ content of amino acids (e.g., tryptophan and lysine)… should be rewritten as follows :”For instance, feeding bees with sugars causes significant alterations in amino acid content of certain amino acids in RJ for example, tryptophan and lysine.

As the reviewer suggested, we have rewritten that sentence to make it clearer (line 485-486).

Line 499: “The exceptional cognitive abilities of queen bees are associated with persistent expression of Dnmt3gene and DNA methylation…”. The sense of this sentence is not completely clear. Should be corrected as follows: ”The exceptional cognitive abilities of queen bees are associated with persistent expression of Dnmt3 gene encoding DNA methyltransferase – enzyme catalyzing DNA methylation”.

Yes, the sentence has been modified as the reviewer indicated (line 504).

Line 544-545 “mitochondrial-related apoptosis [24, 67, 116]” The author need to add the following reference here: ”Effect of g-synuclein silencing on apoptotic pathways in retinal ganglion cells. J Biol Chem, 2008, 284, (52): 36377-85.

Yes, the indicated reference has been inserted in the indicated location (line 1199).

We thank the reviewer once again for his/her sincere advice. We hope that the comments were properly handled and that the revised version will be suitable for publication.

 Best regards,
